# Continuous moulting by Antarctic krill drives major pulses of carbon export in the north Scotia Sea, Southern Ocean

C. Manno[1,2 ✉], S. Fielding[1], G. Stowasser [1], E. J. Murphy[1], S. E. Thorpe [1] & G. A. Tarling [1,2]

Antarctic krill play an important role in biogeochemical cycles and can potentially generate high-particulate organic carbon (POC) fluxes to the deep ocean. They also have an unusual trait of moulting continuously throughout their life-cycle. We determine the krill seasonal contribution to POC flux in terms of faecal pellets (FP), exuviae and carcasses from sediment trap samples collected in the Southern Ocean. We found that krill moulting generated an exuviae flux of similar order to that of FP, together accounting for 87% of an annual POC flux ($22.8\ \mathrm{g\ m^{-2}\ y^{-1}}$). Using an inverse modelling approach, we determined the krill population size necessary to generate this flux peaked at $261\ \mathrm{g\ m^{-2}}$. This study shows the important role of krill exuviae as a vector for POC flux. Since krill moulting cycle depends on temperature, our results highlight the sensitivity of POC flux to rapid regional environmental change.

[1] British Antarctic Survey, Natural Environment Research Council, Cambridge CB3 0ET, UK. [2]These authors contributed equally: C. Manno, G. A. Tarling. ✉email: clanno@bas.ac.uk

Antarctic krill (*Euphausia superba*, henceforth referred to as krill) is highly abundant and a crucial link between primary production and upper trophic levels in the Southern Ocean food web[1,2]. In this region, krill are a fundamental conduit for biogeochemical processes, including nutrient recycling, benthic-pelagic coupling, and carbon (C) sequestration[3,4], and can have an important role in regulating the magnitude of carbon stored in the ocean via the biological pump (BCP) (i.e., the process that draws down atmospheric carbon dioxide ($CO_2$) through the fixation of inorganic carbon by photosynthesis and the consequent export and sequestration of C to the deep ocean)[5]. Krill faecal pellets (FPs), which sink at speeds of hundreds of metres per day, provide pulses of C that can dominate particulate organic carbon (POC) export and the efficient transfer of C through the mesopelagic[6–10]. Recently, Belcher et al.[11] estimated that the seasonal flux of krill FPs within the marginal ice zone of the Southern Ocean is equivalent to up to 61% of satellite-derived estimates of total C flux.

Sinking carcasses of krill can further contribute to the oceanic carbon export (especially outside the phytoplankton growth seasons), becoming a major food source for the benthos[12]. Carcasses result from predation and/or non-predatory mortality such as senescence and starvation, as well as the presence of a number of both external and internal parasites[13,14].

Antarctic krill are a member of the order Euphausiacea, which are unusual among crustaceans in that they continue to moult at regular intervals throughout their adult life[15]. On reaching adult size, most other crustaceans reach a terminal moult or moult irregularly at seasonal transitions or the start of reproductive periods[16]. The recurrent moulting of krill generates a large amount of exuviae with a C content varying between 10 to 23% of total body dry weight[17,18]. Chitin, a polysaccharide that can be completely remineralized to become a source of dissolved organic C, comprises 13% of exuviae[19]. However, despite this potentially large amount of C being discarded into the water column, the contribution of krill exuviae to C export in the Southern Ocean has yet to be adequately considered[20]. The export of C via exuviae has the potential to be highly efficient as krill aggregate in very large swarms[21,22] and rapidly generate large amounts of discarded matter that can over-saturate scavenging communities[11].

Here, we determine the seasonal contribution of krill to the flux of POC in terms of FP, exuviae and carcasses from samples collected by a moored sediment trap deployed for 1 year on the shelf of South Georgia in the north Scotia Sea, in the Southwest Atlantic sector of the Southern Ocean. This sector holds >50% of the circumpolar stock of Antarctic krill[23] and is the geographic focus for the krill fishing industry[24].

Recent interest in the role of krill in Southern Ocean biogeochemical cycles has highlighted the need for better parameterisations of C flux generated by krill[25]. The approach to date has been to consider rates of egestion and respiration multiplied by regional krill biomass estimates[11,26]. Independent validation of these estimates has so far been lacking. To address this, we take an inverse modelling approach to determine the seasonal population size necessary to generate the C flux attributed to krill exuviae. This is made possible from previous detailed parameterisations of krill moulting rates as a function of physiological and environmental parameters[27,28].

This study adds new insights on the important role of krill as a vector for C export in a region that contributes significantly to global atmospheric C uptake[10,29]. We found that, in the north Scotia Sea, krill exuviae can contribute to C flux at a similar magnitude to FPs. Sea-ice decline, ocean warming and other environmental stressors act in concert to modify the abundance, distribution and life-cycle of krill[30,31]. Here, we show the rate of moulting, and release of C-rich exuviae into the environment, is an important contributor to C flux. This must be taken into account when assessing Southern Ocean carbon budget and krill harvesting practices[25].

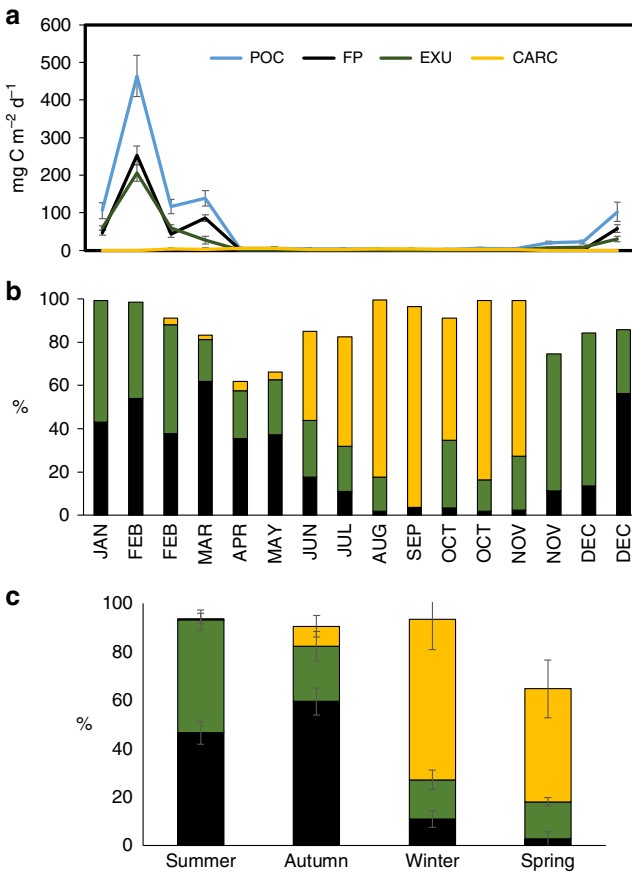

**Fig. 1 Seasonal trend of carbon flux of krill. a** Seasonal particulate organic carbon (POC) flux (mg m$^{-2}$ d$^{-1}$; ±1 SD) and the flux of krill faecal pellets (FP), exuviae and carcasses (mg m$^{-2}$ d$^{-1}$; ±1 SD). Error bars are standard deviations from replicates (sub-samples) from each cup. **b** Relative contribution of krill FP, carcasses and exuviae to total POC. **c** Relative contribution of krill FP, carcasses and exuviae to total POC seasonal average (%, ±1 SD).

## Results

**Contribution of krill (as FP, exuviae and carcasses) to the POC flux.** Particulate organic carbon (POC) flux showed a strong temporal variability, with values ranging from 2.54 to 463.27 mg C m$^{-2}$ d$^{-1}$ (Fig. 1a). POC flux was generally high up to the beginning of autumn (March), after which there was a large decrease (by 2 orders of magnitude) to a low winter level (April to September) until it increased again in the late spring (November onwards). The relative contribution of the different krill components to total POC flux varied between seasons (Kruskal–Wallis one-way analysis of variance (ANOVA), FP, $H = 11.35$, $p = 0.039$; Exuviae $H = 9.33$ $p = 0.025$; Carcasses $H = 8.60$ $p = 0.035$) (Fig. 1b). High POC fluxes corresponded to periods when krill exuviae and FPs were the dominant contributors (accounting for up to 99.2% of which 52% were FPs and 47.2% exuviae). Only when POC flux was low did carcasses become the dominant contributor (up to 82.0% of total POC flux). From a seasonal perspective, the greatest contribution to POC flux was from krill FPs in autumn (59.5%), exuviae in

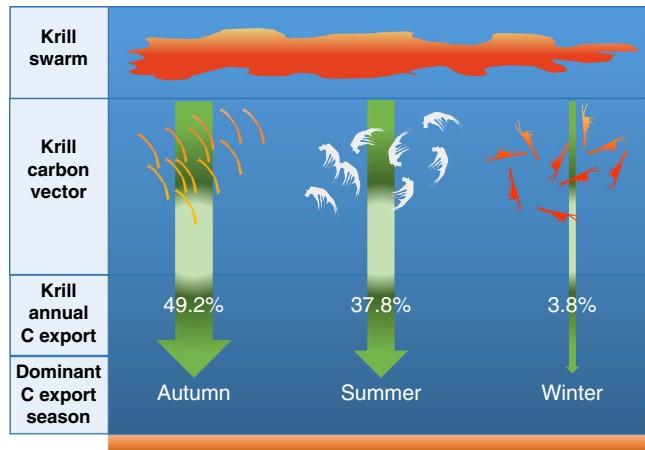

**Fig. 2 Dominant processes driving krill carbon flux.** Schematic showing the average annual proportional contribution (%) of krill derived carbon (faecal pellets, exuviae and carcasses respectively) to annual particulate organic carbon (POC) flux and the dominant export season for each vector.

summer (46.5%) and carcasses in winter (66.4%) (Fig. 1c). Out of an annual total POC flux of 22.8 g C m$^{-2}$ y$^{-1}$(SD ± 4.16), FPs made the highest contribution (49.2%), followed by exuviae (37.8%) and carcasses (3.8%) (Fig. 2) (see Supplementary Table 1.1 for POC flux data).

**Estimate of krill standing stock.** Estimated krill standard lengths were between a minimum of 15 mm and a maximum of 44 mm over the study period, with a median body length across all months of 30 mm (Fig. 3). No significant differences were found in the distribution of standard lengths between months with the exception of March, where median standard length was 24 mm and October, where it was 41 mm, although exuviae numbers collected in the latter month were relatively low (Kruskal–Wallis one-way ANOVA on ranks, $H = 102.78$, 14 df, $p < 0.001$, Dunn's Method post-hoc test, March vs. October, $Q = 4.066$, $p = 0.005$). Standard lengths between 15 and 44 mm are typical of juvenile and sub-adult krill in this region, although smaller mature adults can also fall within this size range[32]. The decrease in standard lengths in March probably resulted from the influx of younger, smaller individuals (15 to 35 mm) into the study region. Uropod length and krill standard length distributions are provided in the Supplementary Table 1.2.

We estimate the average standing stock biomass of krill to vary from a high of 260.9 (SD ± 147.8) g WW (wet weight) m$^{-2}$ in summer to a low of 0.8 (SD ± 0.6) g WW m$^{-2}$ in winter (excluding September, for which there were no captured exuviae; Fig. 4). This calculation assumes that krill were evenly distributed over the daily spatial scale of capture (i.e., 1 to 2 km, representing the horizontal distance travelled by a passively moving exuvia before reaching the sediment trap at 300 m depth, as determined from simultaneous current velocities from an acoustic Doppler current profiler, see Supplementary Methods). To bracket these values to account for extreme scenarios of patchiness over this scale, we used coefficient of variation (CV) estimates for krill biomass distribution from Fielding et al.[33]. That study performed mesoscale acoustic surveys over the north-western shelf of South Georgia, encompassing the site of the present sediment trap and estimated CVs for spatial cells of 500 m length along the acoustic transects. We consider those surveys to provide a reasonable estimation of distribution variance over the 1 to 2 km present spatial scale of capture, although we note their study was inter-annual (16 years) and not seasonal. Across all surveys in Fielding et al.[33], the 95% percentile CV value was 58.6%. For the peak

seasonal value of 260.9 g WW m$^{-2}$, this generates an upper bound of 413.8 (SD ± 234.4) g WW m$^{-2}$ and a lower bound of 108.0 (SD ± 61.1) g WW m$^{-2}$. For the lowest seasonal value of 0.8 g WW m$^{-2}$, the upper bound is 1.2 (SD ± 1.0) g WW m$^{-2}$ and the lower bound, 0.3 (SD ± 0.3) g WW m$^{-2}$. Krill biomass data are provided in the Supplementary Table 1.3. A detailed explanation of these calculations are provided in Supplementary Methods.

## Discussion

We found that krill can be a dominant contributor to POC flux in the north Scotia Sea (specifically on the South Georgia shelf). Their contribution is mainly generated from their faecal pellets and exuviae, which, together, comprised 92% of annual total POC export in the present study region. At its seasonal peak, the contribution of krill can substantially augment the total flux of POC to levels in excess of 460 mg m$^{-2}$ d$^{-1}$, which is an order of magnitude greater than that observed even in highly productive, iron-fertilised regions within the Southern Ocean (POC flux up to 23–27 mg m$^{-2}$ d$^{-1}$)[10,34] and more similar to POC values observed in other high krill density regions such as the Bransfield Strait[35]. Hence, we suggest that some of the strongest carbon sinks in the Southern Ocean occur in regions where both high primary productivity and high krill concentrations coincide.

The important role of krill FPs in promoting C flux has already been highlighted by Belcher et al.[11] who calculated the pulse of C generated by krill FPs in the marginal ice zone. However, the present study shows the almost equal contribution of krill exuviae to C export in the Southern Ocean. This indicates that the initial estimates of Belcher et al.[11] could be almost double when the moult cycle is taken into account. Furthermore, the large contributions of krill FPs and exuviae to C flux, across the full summer and early autumn, may be an important source of nutrition to fuel the benthos through the winter.

We observed a large decrease in C flux generated by krill at the beginning of winter, which persisted until the following spring. This may be, in part, a result of a reduction in population biomass, as previously observed in this region in March by Saunders et al.[36]. Large aggregations of krill must occur during winter in most years on the South Georgia shelf as that is when the krill fishery occurs. For instance, 18,558 tonnes were taken from the region during winter in 2017[37,38]. However, although the fishery has historically operated near the present study site (north-west South Georgia shelf), it has mainly concentrated in the eastern South Georgia shelf region during the last decade. The decrease in C flux may also be the result of a decrease in physiological activity during winter[39] resulting in a seasonal reduction in feeding[40], growth rates[27,41,42] and depression in moulting rate[40].

Decrease of physical retention of krill in shelf waters in autumn[43] can also potentially lead to a reduction in krill concentration and C flux. Alternatively, the reduction in C flux may be a result of krill migrating below the sediment trap during winter. The vertical distribution of krill has been reported to deepen during winter[32,44,45], and in situ observations of individual behaviour have revealed a net pattern of downward swimming at the end of the productive season[46]. However, the sediment trap in the present study was relatively close to the seabed at 300 m, implying that krill must be resident close to the seabed to be below the trap. Although it is known that krill do interact with benthic sediments[47], it is unknown whether they maintain an epibenthic distribution for many months. There also remains the possibility of a large-scale shift in the distribution of krill out of the study area, but there is little evidence to support this view. It has been suggested that major movements of krill to more inshore regions occur during the winter at the Antarctic

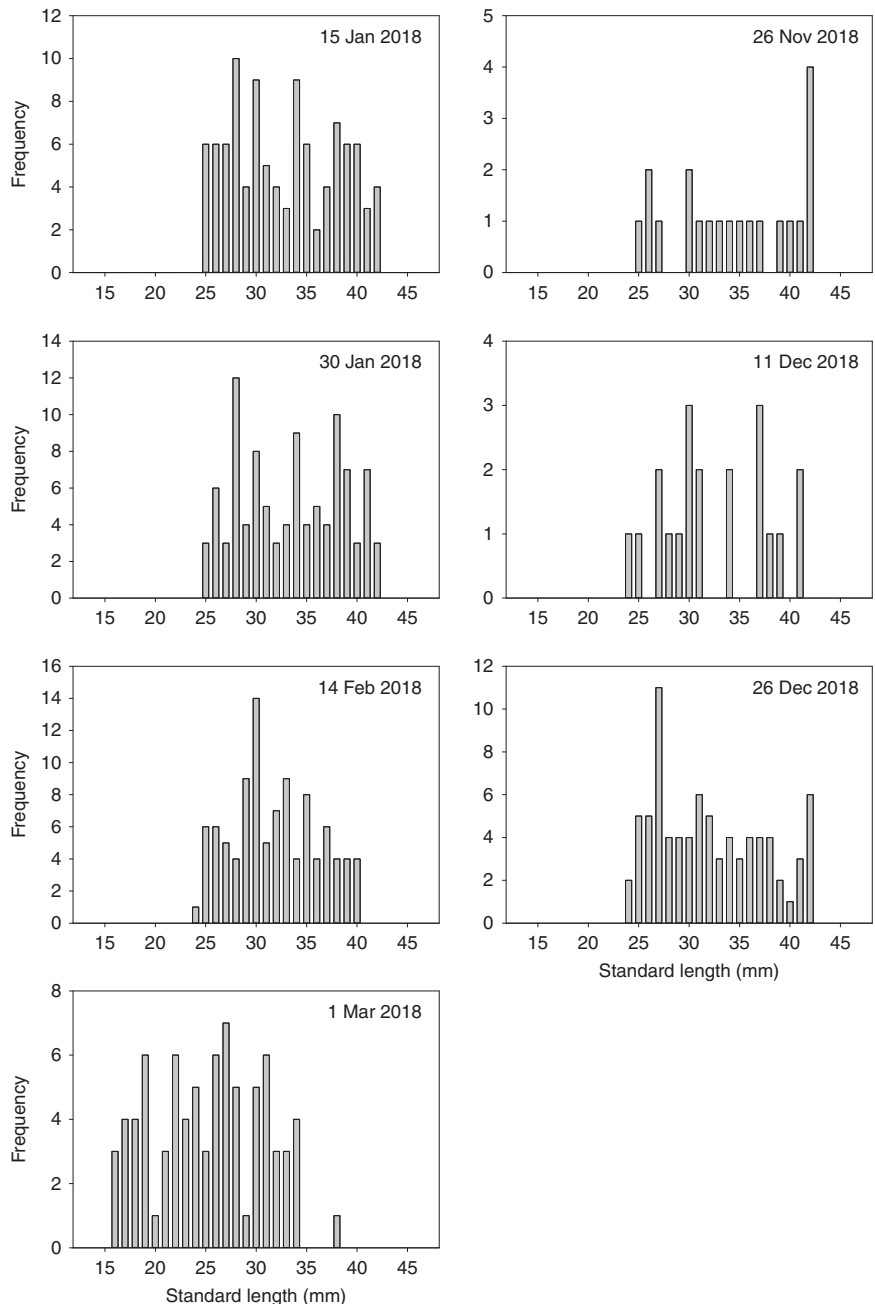

**Fig. 3 Krill length frequency.** Population frequency of krill standard lengths, based on uropod lengths of exuviae caught in the sediment trap. Note: only months with a significant number of exuviae were selected for the calculation. Subplots are labelled according to the date the sediment trap bottle opened.

Peninsula[45]. Further, Reiss et al.[48] observed an order of magnitude increase in krill biomass in the Bransfield Strait in winter, suggesting that this increase must have resulted from active horizontal migration from offshore areas occupied during summer. Nevertheless, evidence is presently lacking for similar migrations elsewhere. Overall, we suggest that the large decrease in exuviae and FP flux in the winter is likely due to a combination of the biological and physical processes mentioned above.

Assuming that krill were evenly distributed in the vicinity of the sediment trap, we estimate that the krill standing stock required to generate the observed exuviae flux was between 1 and 261 g WW m$^{-2}$. These values are of the same order of those estimated by the mesoscale acoustic and scientific net study of Fielding et al.[33], of between 3 to 137 g WW m$^{-2}$, for the same region over a 20-year time series. The consistency of these estimates is reassuring

given that they were derived from completely independent methods. It further places some confidence in using moored devices to derive krill density estimates, particularly in being able to use exuviae from sediment traps to determine both population structure and abundance that can complement estimates of biomass from autonomous active acoustic devices[49]. The sediment trap estimate nevertheless assumes that the distribution of krill over wider spatial scales is the same as within its zone of capture, and that patchiness does not cause any extreme bias in the numbers of captured exuviae over the 15 to 30 day collection period. This requires further verification, which could be achieved through analysis of acoustic surveys in the vicinity of the mooring. The combination of acoustic technology with autonomous collection devices has a great deal of potential to obtain data from regions that are remote and difficult to access, particularly during winter.

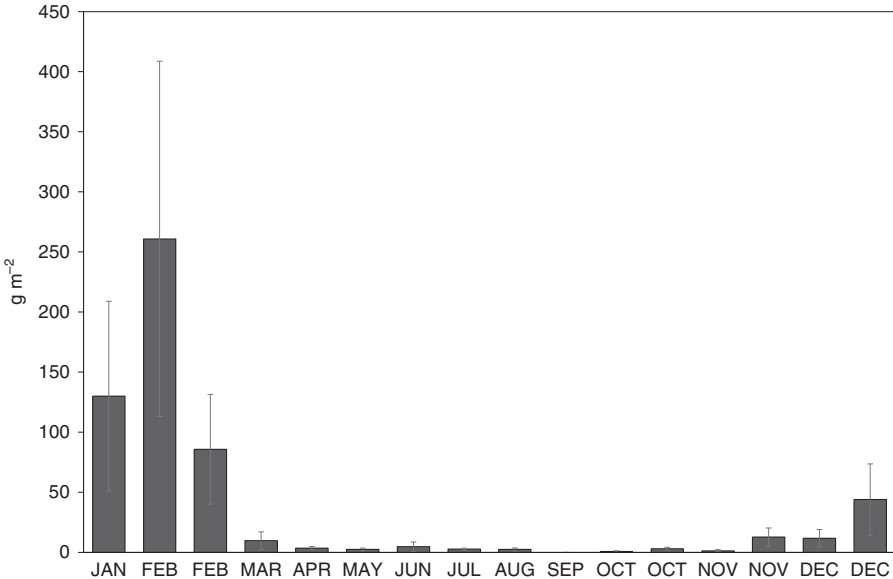

**Fig. 4 Biomass of krill population.** Mean biomass (wet weight) of krill (g m$^{-2}$, ±1 SD) from each sediment trap bottle period based on an inverse model calculation based on exuviae caught in the sediment trap.

After spawning, krill eggs can sink to depths in excess of 1000 m before hatching and returning to the surface to feed[50,51]. Neither eggs nor larval stages of krill were found in our sediment trap samples at any time of year, indicating an absence of successful spawning in this region. We did observe a decrease in the size of exuviae within the sediment traps during autumn (March), indicating a recruitment of juvenile krill into the study region. These krill were juvenile (15+ mm) and are most likely to have originated from areas upstream of South Georgia, such as the Antarctic Peninsula and outlying islands[33,52,53]. Interestingly, this increase in juveniles in March also resulted in a secondary peak in FP flux, indicating high levels of feeding activity and egestion, likely supporting rapid levels of growth in juvenile stages, as observed by Tarling et al.[27] and Atkinson et al.[41].

The present study illustrates the dominant role that krill can play in driving the sinking flux of POC in the Southern Ocean. As well as supporting the findings of other studies regarding the contribution of FPs[11,54,55], it further identifies the major contributions made by exuviae and sinking carcasses. The process of regular moulting by krill can generate a flux almost equal to that of the FP flux. Assuming a krill population biomass of 379 Mt for the Southern Ocean[56], we estimate that the exuviae flux can contribute a seasonally averaged mean of 0.29 Mt C d$^{-1}$ (SE 0.09; see Supplementary Methods). As well as being a major source of sequestration, this flux can also be a major driver of productivity in benthic communities, particularly in shelf regions.

Increasing water temperature in the Scotia Sea, as a result of climate change, will likely have a negative impact on krill growth and biomass[57,58]. Here, we show for the first time the crucial role of krill exuviae as a vector for C flux in the Southern Ocean, a region which contributes significantly to the global C export production[59]. Thus, a potential decrease in krill biomass is likely to impact the marine biogeochemical cycles. Further, since the krill moult cycle (and in turn exuviae production) depends on temperature[28], our findings highlight the sensitivity of C flux to rapid regional environmental change[30,31].

## Methods

A bottom-tethered mooring was deployed at a single site (WCB mooring platform) for approximately 12 months between January–December 2017. The mooring was located at 53° 47.90'S, 37° 55.99'W and deployed at 300 m on the South Georgia shelf (north Scotia Sea) (Fig. 5). The sediment trap (McLane Parflux sediment traps, 0.5 m$^2$ surface collecting area; McLane Labs, Falmouth, MA, USA) carried 21 receiving cups and was fitted with a plastic baffle mounted in the opening to prevent large organisms from entering. Each bottle contained a solution of 4% formalin mixed with 5 g Sodium Tetraborate (BORAX) to arrest biological degradation during sample collection and to avoid carbonate dissolution. The sample carousel was programmed to rotate at intervals of 15 days in austral summer and 30 days in austral winter. Data on the hydrographic conditions at the mooring site over the duration of the deployment were acquired by an acoustic Doppler current profiler (ADCP) operating at 300 kHz, and a Seabird SBE37 conductivity/temperature/depth logger (CTD), of which both were deployed on the main mooring buoy at 187 m.

**Trap sample processing and analyses.** Once in the laboratory, the supernatant of each cup was removed by pipette and its pH was measured. Prior to splitting, swimmers (i.e., zooplankton that can enter the receiving cups while alive) were carefully removed: samples were first wet-sieved through a 1 mm nylon mesh and the remaining swimmers hand-picked under a dissecting microscope. Large aggregates, empty tests and exuviae retained by the mesh were returned to the sample. Each sample was then divided into a series of replicate fractions (pseudo-replicates) for subsequent analysis using a McLane rotary sample splitter (McLane Labs, Falmouth, MA, USA).

**POC analysis.** Replicate fractions were vacuum filtered through pre-weighed and pre-combusted Whatman GF/F filter (550C for 5 h) for Particulate Organic Carbon (POC) analysis. Filters were then desalted by a short wash with distilled water and dried at 60 °C. POC was measured by combustion in an elemental analyser (CHN); for POC determination, filters were previously treated with 2N H$_3$PO$_4$ and 1N HCl. POC flux was expressed in mg m$^{-2}$ d$^{-1}$, estimated by dividing the total mass per sample by the time interval and the trap collection area. Seasonal flux was averaged for each season as follows: bottle 1-2-3-14-15-16 (summer, from Dec to Feb), bottle 4-5-6 (autumn, from March to May), bottle 7-8-9 (winter, from June to Sept), bottle 10-11-12-13 (spring, from Sept to Nov).

**Faecal pellets analysis.** Krill FPs were counted under light microscopy. The dimensions of krill faecal pellets were measured (length and width) using an ocular micrometre, from which pellet volume was calculated by the geometric formulae associated with the cylindrical krill FP shape[60]. To estimate the contribution of the FP to POC, the carbon content of FPs was calculated using a seasonal conversion factor specific to the Scotia Sea[10]. These seasonal conversion factors used estimates of the carbon content of euphausiid FPs from sediment traps deployed at oceanic stations upstream and downstream of South Georgia. Seasonal conversion factors were derived for two periods, late spring to early autumn (October to April, 0.030 mg C mm$^{-3}$) and late autumn to the end of winter (May to September, 0.018 mg C mm$^{-3}$), based on differences in the C content of the krill FPs that reflected a change in food sources[10].

**Exuvia and carcass analysis.** Exuviae and carcasses of krill were picked out under a light microscope and their dimensions measured. They were then carefully rinsed and dried at 60 °C for 24 h before their dry weight (DW) was measured. The sum

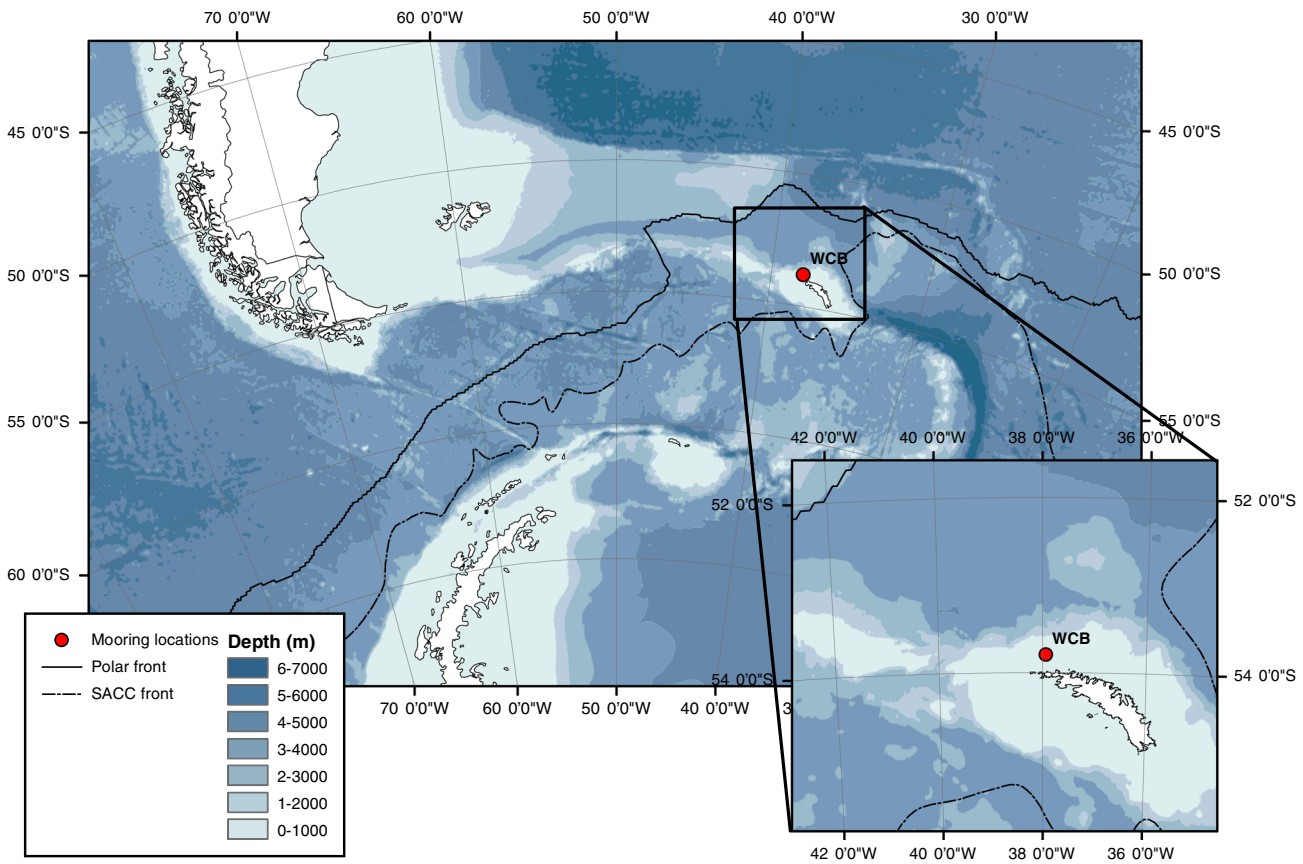

**Fig. 5 Location map Location of sediment trap mooring (WCB) in the Scotia Sea, Southern Ocean.** Bathymetry data from Gebco. Map generated using arcmap 10.6.

total DW of exuviae and carcasses for each sample was calculated. One-hundred exuviae were randomly picked from different samples, measured and then analysed for C content using an elemental (CHN) analyser. The C content of carcasses was calculated using a DW to C conversion provided by Atkinson et al.[2] for Scotia Sea krill (where C = 41.7% of DW).

**Estimation of krill standing stock biomass.** The estimation of Antarctic krill biomass (g WW [wet weight] m$^{-2}$) was carried out through 5 major steps: (1) determination of the intermoult period (IMP), which also represents the number of randomly moulting krill required to produce one exuviae per day; (2) calculation of the equivalent wet weight (WW) of a moulting krill; (3) estimation of the spatial scale of capture and the dimensions of the sediment trap capture zone; (4) combination of parameters 1 to 3 to derive mean krill biomass; and 5. derivation of reasonable upper and lower bounds for the mean biomass estimate. Full details of these calculations are provided in the Supplementary Methods and are only briefly summarised here.

Krill total length was estimated from uropod length following Miller[61]:

$$S1_{i,j} = 8.192 + 5.233 UL_{i,j}, \tag{1}$$

where S1 is estimated standard body length (mm) and UL is measured uropod length (mm) from an exuvia, $i$ is the specimen and $j$ is the sediment trap bottle. The number of exuviae measured from each bottle ($m_j$) was a maximum of 100. It is to be noted that S1 actually represents the length of exuvia and not of the live krill, which may have grown or shrunk subsequent to moulting. Nevertheless, given that growth increments at moult are comparatively small, and may either be positive or negative[26,41], we assumed exuvia length to be a reasonable estimate of krill length.

For the majority of bottles, $m_j$ encompassed all of the exuvaie collected ($n_j$), but was less than $n_j$ in some others, which reached up to a maximum 280 exuviae. The IMP per individual, $IMP_{i,j}$, was determined through applying equations provided by Tarling et al.[26], and extracting climatological sea surface temperature from Whitehouse et al.[62]. For the winter period (March to August), $\overline{IMP}_{i,j}$ was multiplied by 5, in line with the increase in winter IMP reported by Buchholz et al.[40]. It is not possible to determine sex/developmental stage from exuviae, so IMP was estimated for all possible stages and an average IMP across these stages, $\overline{IMP}_j$, subsequently derived. The required number of krill to generate the average daily number of captured exuviae ($e_j$)was subsequently $\overline{IMP}_j \cdot e_j$. This assumes that all krill in a population moult randomly in relation to each other, which has

been validated by histological studies on the moult cycle by Buchholz et al.[40] and observational studies on incubated specimens by Tarling et al.[26].

The WW ($w$, g) of an individual krill, $i$, caught within sediment trap bottle $j$, was estimated using the following equation, provided by Kils[63]:

$$w_{i,j} = 1.58 \cdot 10^{-6} \cdot S1_{i,j} \tag{2}$$

We define the term spatial scale of capture as the distance between the sediment trap and the remote location from which a captured exuvia was released. Within this scale, all moulted exuvia have the potential to be captured by the sediment trap. Not all potential captures will be realised, however, since a number of forces act on the sinking exuvia, moving them in and out of the capture zone, i.e., the zone which figuratively lies within the mouth area of the sediment trap (Supplementary Fig. 2.1). How representative the number of captured exuvia are will depend on the distribution of krill within the spatial scale of capture (i.e., even or uneven) and the capture period over which the results are integrated. For the present mean biomass estimate, we assume an even distribution based on the principle that the lengthy sampling interval (15 or 30 days) will moderate any spatial unevenness in krill distribution within the spatial scale of capture.

To estimate the spatial scale of capture, we used simultaneous current velocity data from a moored acoustic Doppler current profiler moored with the present sediment trap. Further, we assumed that (i) all krill were located in the top 50 m of the water column during night-time[22] (ii) the sinking speed of an exuvia was 1 cm.s$^{-1}$ [20], such that the maximum time taken to reach the sediment trap at 300 m depth was 8.3 h, (iii) the exuvia moved passively with prevailing horizontal ocean currents as it sank through the water column.

To estimate mean krill biomass ($B_j$, g WW m$^{-2}$), we firstly calculated the biomass of krill required to generate $e_j$ (i.e., average daily number of captured exuviae) as follows:

$$G_j = \frac{\sum_{i=1}^{i=m} \overline{IMP}_{i,j} \cdot w_{i,j}}{m_j} \cdot e_j \tag{3}$$

Assuming that the number of krill within the capture zone is representative of that over the spatial scale of capture (i.e., an even distribution), the sampling volume ($V$) is accordingly the mouth-opening area of the sediment trap (0.5 m$^2$) multiplied by the water depth over which moults were released (50 m). $B_j$ (m$^{-3}$)

then becomes:

$$B_j = G_j \cdot V^{-1} \qquad (4)$$

$B_j$ values were converted into units of $m^{-2}$ through multiplying by 50 m. Standard deviation (SD) values were calculated using stage-specific IMP data for each krill.

Unevenness in krill distribution is a potential source of variability in exuvia capture rates. We accounted for this through deriving upper and lower bounds for mean biomass based on estimates of krill distribution variability for this same region by Fielding et al.[33]. Fielding et al.[33] carried out acoustic surveys, which were analysed through dividing transects into cells of 500 m spatial length. We considered this length scale appropriate in relation to the present spatial scale of capture of 1 to 2 km. Fielding et al.[33] derived coefficients of variation (CV) between cells for 16 different surveys, for which we extracted the 95% percentile value (58.6%). Upper and lower bounds for the mean biomass for each sampling interval $j$ were estimated by inflating or deflating $e_j$ by this value. SD values were calculated as for the mean biomass estimate.

**Statistical analysis**. Kruskal–Wallis one-way ANOVA H-Test was used to determine whether there were any significant differences between seasons with regard to POC, FP, carcasses and exuviae. The same test was used to determine any differences in the distribution of standard lengths between 15 or 30 day collection periods (results reported above). Differences were considered significant where $p < 0.05$.

## Data availability
All data are available at https://doi.org/10.5285/60F9B354-2CBF-4D4D-84F7-13C9224F82D4 and presented in the supplementary material.

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

## Acknowledgements

We thank the captain, officers and the crew of the *RRS James Clark Ross* and Discovery for their support in all the logistical operations on board, and we also thank the Antarctic Logistics and Infrastructure programme at British Antarctic Survey, and Peter Enderlein and Bjørg Apeland for support of the deployment and recovery of the mooring. We thank Dr. Gérald Darnis for the discussion on exuvia export. This work was carried out as part of the Ecosystems programme at the British Antarctic Survey and the Scotia Sea Open Ocean Laboratories (SCOOBIES) sustained observation programme at the British Antarctic Survey in the frame of WCB-POETS survey cruises.

## Author contributions

C.M. conceptualised the project, analysed the samples and wrote the manuscript with the assistance of all authors. S.F., G.S. and C.M. carried out the fieldwork. S.F. processed ADCP data, G.T.A. developed the calculation to estimate krill population size and S.T.E. provided the oceanographic data analysis support. E.M.J., S.F., G.T.A. and G.S. provided overviews on krill biology and particle flux.

## Competing interests

The authors declare no competing interests.
