## [Peer Review File · Nature Communications]

REVIEWER COMMENTS

Reviewer #1 (Remarks to the Author):

Review of Manno et al.

The manuscript entitled "Continuous moulting by Antarctic krill drives major pulses of carbon export in the Scotia Sea, Southern Ocean" uses sediment trap samples to assess the contribution of the flux of krill exuviae, fecal pellets, and carcasses to total particular organic carbon (POC) flux. It sheds important light on the role of Antarctic krill in mediating carbon transport and sequestration and shows that carbon flux through sinking krill exuviae is nearly equal to that of their sinking fecal pellets. This is an important finding as it increases our understanding of the Southern Ocean as a carbon sink. I think that the use here of sediment trap samples to demonstrate the importance of exuviae and krill moulting to carbon flux is strong. I have some reservations, however, with the approach taken to estimate krill biomass and abundance from the exuviae found in the sediment trap cups (it could just be how the approach has been explained). I provide more details on this below. I have additional concerns over other aspects of the methods too that I describe below. On the whole, I think the manuscript is well written, novel, and relevant to the scientific community. I recommend that the manuscript be reconsidered after major revision.

Specific Comments:

Lines 32-38: This paragraph doesn't flow well, and I would recommend revising it for improved readability.

Line 33: What do you mean by "predator activity"? Is this just predation?

Line 56: Remove "krill" after "circumpolar" so that it doesn't read "krill stock of ...krill".

Line 81: I think "varied" is better here than "altered".

Line 85: What months did you use to get averages of each season?

Line 91: Did you use a statistical approach to test whether the range in lengths was similar throughout the study period or was this just eyeballed? It seems it was just eyeballed, but it would be better to statistically test this.

Line 93: It is certainly possible to have mature adults that are within this size range though.

Lines 99-100: I think you mean the other way around here – "maximum and minimum levels of biomass occurring in the summer and winter, respectively".

Lines 100-101: I'm not sure I agree with this approach. It is possible that the exuviae you observed in the winter came from the same biomass of krill estimated from the summer. I don't think summing the biomass across seasons is a good approach.

Line 103: Add a period/full stop at the end of this sentence.

Line 110: Needs rephrasing – I suggest addition of words in bold "which is an order of magnitude greater than that observed even in highly productive...".

Line 122: Why "must" this "outlast the preceding winter"? And can you show that with your data? Do you rather just mean that it is likely sufficient to fuel the benthos through the winter? And if so, what references can you provide to support this?

Lines 127-128: Doesn't the krill fishery start in the summer? Can you say that it extends into the winter, therefore large aggregations must, in most years, persist into the winter in the region?

Line 133: The citation here seems odd having "- Siegel book chapter" tagged onto it. If this is the way Nature Comms requires book chapter citations, then disregard this comment, but if not, then correct the citation please.

Line 137: Either "lead to a reduction in" or "lead to reductions in", but not as currently written.

Line 149: I believe the work of Reiss and colleagues demonstrates the move of krill biomass inshore into the Bransfield Strait region in winter. Order of magnitude higher biomass in this region in winter compared to summer.

Lines 150-152: A little awkwardly phrased. I suggest "We suggest that the large decrease in exuviae and FP flux in the winter is likely due to a combination of the biological and physical processes mentioned above."

Lines 165-166: Is this surprising given your sediment trap was at 300m and female krill typically only release eggs over waters that are >1000m?

Line 168: The way this is phrased gives the impression that the juveniles developed from eggs spawned that same season, which would not have been the case as they first must survive a winter. Do you rather mean that these were larval krill? The size range suggests at least some of them may have been. And that would make more sense.

Lines 168-171: I'm not convinced that the evidence provided here supports allochthonous recruitment.

Line 171: Typically, the new recruits appear in Spring after overwintering as larvae not in autumn on the same year they were spawned.

Line 192: Which months were classified as summer and which were winter?

Line 196: Should be "these data" rather than "this data" since data is the plural of datum.

Line 233: "Krill total length" should rather be "Exuviae total length" shouldn't it? Since the length has been estimated from the exuviae uropod rather than a live krill uropod. It is also necessary to include the caveats of doing this – i.e. that you aren't actually getting length measurements of live krill, but rather of their exuviae.

Line 236: I believe it should be SL for Standard Length, rather than S1.

Lines 238-240: As for comment above – SL not S1.

Line 239: "from exuviae (v)" – v is not in the equation but is provided as if it is.

Line 240. Sentence is missing a period/full stop.

Lines 252-253: Why would it be assumed that they moult randomly to each other? If there is a citation for this, please provide it here. And, by stating this, are you also saying that only one moults on a given day? If so, this is definitely not the case.

Lines 253-256: IMP is the number of days between moults for an individual krill, and 1/IMP is the number of moults in a given day for that same individual (and would be a fraction of 1 unless they're moulting more than once a day). I don't think I've ever seen 1/IMP being referred to as the number of krill that moult on any given day. I don't see where this jump in logic comes from. How can the inverse of the inter moult period, the number of days between moults, become the number of krill that moult in one day? Perhaps I'm missing something here, but that would suggest that this section needs to be revised for clarity.

Lines 267-269: I'm not sure why this is mentioned here as you don't present the data in this format.

Line 269: Period/full stop missing from the end of the sentence.

Line 272: Please clarify how seasons were allocated somewhere in this manuscript.

Line 273: Just a quick look at the data for POC, exuviae, and FP especially (maybe not as much for carcasses), suggests that the data are not normally distributed. Also, I used the data you provided in the Supplemental Materials and looked at normality myself, and the data are very clearly skewed. Can you provide the p-values for these tests for normality and equal variance that were passed please? Or maybe they were transformed already?

Line 508: Period/full stop missing before "c)".

Line 509: Period/full stop missing at the end of the sentence.

Line 520: Period/full stop missing at the end of the sentence.

Line 521: Period/full stop missing at the end of the sentence.

Figure 2: Looks quite strange with the error bars overlapping like that – can you have grouped bars rather than stacked?

Supplementary Data – why are their asterisks in the units? E.g. $\text{mg}\cdot\text{m}^{-2}\cdot\text{d}^{-1}$ should just be $\text{mg m}^{-2} \text{d}^{-1}$

Supplementary Information – I think this map figure would be more useful in the manuscript itself.

More detailed comments/questions on methods:

1. Were the trap samples preserved in formalin in situ? If not, how did you account for bacterial degradation? If they were preserved in formalin in situ, is it not possible that a krill could swim into the cup and become preserved, rendering it a "carcass"? There is no mention of preservation, but I know from other work that this is typically done. Please provide the details needed.
2. What were the temperatures for each time period that were used in the calculations of IMP? Please

provide these in your manuscript or supplemental materials.

3. How were the "winter" and "summer" months allocated for the 15 and 30-day time periods? E.g. was April considered winter?

4. Supplementary Data 1.3 – My understanding from the Methods section is that one biomass was calculated for each time period. If that is true, how could there be an SD associated with the value? If this is not true, and biomass shown here is an average for multiple days of the time period, how was this calculated? If you took sub-samples from each cup, that would be considered pseudo-replication.

5. I tested out the equations given in the Methods section to estimate krill biomass for April given the available uropod lengths provided in the supplementary materials. I assumed since only 4 were measured, that there were only 4 moults in the trap cup for April. I had to guess at the temperature (please provide those for each month) and had to assume that April was part of "winter" and therefore the time period was 30 days. I applied the equation provided to convert uropod lengths to standard length, and then averaged the standard lengths (30.75mm). I then use Tarling's equations to calculate the moult probability for juveniles, males, immature females, and mature females for that average SL and a guessed temperature of 1 °C, and then calculated the average moult probability from that – it was 0.33. Then, I calculated the average IMP for April as $30/0.29=91.10$ days. I calculated the value for V (integrated krill) as $4/0.5m^2/30$ days to get 0.267 ind./m²/day. With this, krill abundances, K, were calculated as the average IMP-1 x V to get ~24 ind./m². I applied the equation to calculate biomass, W, and reached a biomass of 0.001 g/m². When I compared this to the value provided in the Supplementary Materials for April, it was very different (3.52 g/m² is provided there). I also tried it with different temperatures and with the period set to 15 days, in case I'd gone wrong there. I'm now very confused. Either the methods need to be more clearly explained, or there is an error somewhere.

Reviewed by Professor Kim Bernard, Oregon State University

Reviewer #2 (Remarks to the Author):

The manuscript by Manno et al. find support for previous observations that krill fecal pellets can be a major contributor to carbon export in the Southern Ocean. The novelty of the manuscript is that in addition to krill fecal pellets, krill exuviae from continuous moulting by krill may be an important source of carbon export in the Scotia Sea. This is a mechanism that has so far been overlooked in the material collected in sediment traps. The carbon export by krill exuviae can match the carbon export from krill fecal pellets as observed by Cavan et al. (2015) and Belcher et al. (2019), making krill the dominant source for carbon export in the Scotia Sea.

The manuscript presents an overlooked potential pathway for carbon export in the Southern Ocean and suggests that data sets from long-term moored sediment traps may offer a method to evaluate the impact from krill exuviae for carbon export in future studies, and potentially from past collected samples, if the material is still available.

The manuscript has some issues that need to be addressed. The author measured total POC flux from the sediment trap samples and collected exuviae for POC measurements. Unfortunately the authors used conversion factors to evaluate the role of both krill fecal pellets and carcasses to estimate their contribution to total POC export. It would have been a much stronger data set if the authors had also measured the POC of the fecal pellets and carcasses. Especially the fecal pellet carbon would have improved the manuscript, since those are most likely to change seasonally depending on the food source from which the fecal pellet were produced.

The estimations for the krill standing stock is very hard to follow. The authors refer to previously published equations by Tarling et al. (2006), however, Tarling et al. (2006) is not cited in the reference list, as far as I could determine. I found the Tarling et al. (2006) paper - assuming it to be: "Natural growths rate in Antarctic krill (*Euphausia superba*): I. Improving methodology and predicting

intermolt period" published in *Limnology and Oceanography*: 51(2): 959-972. Only after going carefully through the equations in the current manuscript and the equations in Tarling et al. (2006), it became somewhat clear how the authors had estimated the krill standing stocks. Here there are several assumptions made by the authors of Manno et al. (current manuscript) in order to get the standing stocks. It is not clear to me how the authors addressed one of the major premises presented by Tarling et al. (2006) in order to use the method, where the number of moltings per day must be constant and equal to the inverse of the molt duration. How did the authors ensure this criteria? Another aspect which was not clear to me was how the authors estimated the capture area of the trap? To me it seemed that the authors assumed that all exuviae that were captured by the trap sank vertically without any horizontal movement? Typically a sediment trap collect material from an area larger than its funnel opening, this would make their estimated standing stock considerably smaller than the one estimated in the current manuscript? This is especially related to the method description from line 241 to line 269. Here the authors first calculate the total concentration of krill per square meter above the trap - this seem that the authors assume all krill to be distributed in a water column of one square meter above the trap, surely this area has to increase as you are further away from the sediment trap and closer to sea surface. The next aspect is when the authors calculate the concentration per cubic meter in the upper 50 m of the water column. Here the authors argue that all krill are within the top 50 m of the upper water column, which again may be seasonally dependent, but is a reasonable assumption. Here I would like to see what the "collection funnel" in the water column above the trap was, dependent on the current velocities, and then see the authors provide a range for the krill standing stock and biomass in the water above the trap.

Overall, the authors present an interesting new mechanism, but it is to a large extent based on few actual measurements and many conversions and assumptions. The authors should address the assumptions and discuss their impact on their findings.

The manuscript is well written and I have only found a few issues to the text itself:

Line 32-34 could use a citation to the statement.

Line 38: I guess Tang et al. (2014) was studying copepods and not krill - are there any studies of krill?

The estimations for krill standing stock should be explained more detailed so the reader does not have to go carefully through Tarling et al. (2006) in order to understand what was done in the present manuscript.

Tarling et al. (2006) needs to be included in the reference list.

Assumptions should be better highlighted and discussed (see above).

It is surprising how much of the seasonal flux (even in early spring) was contributed by krill (either as fecal pellets or exuviae). What was the rest of the exported material? Can you perform POC measurements for krill fecal pellets at different seasons?

At times in the manuscript it is unclear if the authors are referring to dry weight or wet weight.

I miss a paragraph in the discussion addressing potential climate change and its role on the observed mechanism? What happens when the Southern Ocean is warming and the habitat of krill will be restricted by warmer water temperatures? Can you learn anything from your seasonal data?

I believe that the authors are having a long time-series of sediment trap data, why only present one year for this mechanism? The authors could have investigated several years, at least for the summer period when krill exuviae seem to play a crucial role for the total exported POC?

Reviewer #3 (Remarks to the Author):

General comments:

This manuscript represents a well-written, useful study analyzing the contribution of Antarctic Krill, *Euphausia superba*, contribution to particulate organic carbon export (POC) in the Scotia Sea of the Southern Ocean. The major findings of the study show that krill exuviae flux is of similar orders of magnitude to that of fecal pellet flux, which indicates previous studies only quantifying krill fecal pellet flux may underestimate the overall contribution of krill to POC flux. The novel aspect is that no study to date has quantified the contribution of krill exuviae to POC flux in the Southern Ocean. In addition, the study utilizes an inverse model to estimate krill population size based on measured exuviae collected in the sediment trap. These data presented in this study will be valuable to researchers interested in Antarctic krill zooplankton, and to those interested in quantifying zooplankton contributions to the biological pump in the Southern Ocean. This study will influence thinking by encouraging other research programs with sediment traps in the Southern Ocean to quantify fecal pellets and exuviae when determining krill contributions to POC flux. While the claims are convincing, some of the broader conclusions are oversold and need to be better qualified or removed. The sensitivity of POC flux to environmental change is not supported by the dataset in this study and the connection that krill play in contributing to atmospheric carbon drawdown needs to be better explained. Overall, the study is sound and answers the call by Cavan et al. 2019, Nat. Commun. to better parameterize Antarctic Krill contributions to biogeochemical cycles.

Specific comments:

ABSTRACT

L7: Use more active language in abstract... "Antarctic krill are important in..." or "Antarctic krill play an important role in..."

L11: Include specific location...in the Scotia Sea of the Southern Ocean.

L15-17: This conclusion is vague. Also, the study does not analyze the effects of environmental conditions on moulting rate so the way the sentence is currently stated is unjustified.

INTRODUCTION

L23: Should it be krill "are" a fundamental conduit?

L24: Character issue throughout MS for benthic-pelagic coupling (as well as in references).

L45-46: Change to, "Chitin, a polysaccharide that can be completely remineralized to become a source of dissolved organic carbon, comprises 13% of exuvia."

L67-68: It's important to note that the Scotia Sea represents an important location for dissolved inorganic carbon drawdown whereas krill fecal pellets and exuviae would be contributing to organic carbon export. Please ensure to clarify this difference in the MS.

L69: What is "this" modifying? This carbon drawdown?

L72-73: Again, I find this connection weak. The study doesn't address the effect of environmental conditions (e.g., temperature) on moulting. Instead, the authors could highlight they quantify the differences in C flux seasonally.

RESULTS

L81-83: Please identify the relative contribution of exuviae and FPs separately to this 99.2% total POC flux.

L88: Add a % symbol next to 37.8.

L99-100: Should this be flipped to minimum and maximum levels of biomass? I would expect maximum biomass to occur in austral summer but the way the sentence is currently stated makes it appear the maximum is in winter.

L103: Period after SM_1.

DISCUSSION

L109: Contribution of krill "carcasses" can...

L132-135: Tie this single sentence paragraph into the paragraph above it (L124-131).

L145-147: What about habitat partitioning? A recent study showed distributions of calyptope and furcilia larvae concentrate offshore from the Scotia Sea whereas juveniles strongly concentrate on the Scotia Sea shelf and these distributions can vary seasonally (see Perry et al. 2019 PLoS ONE). This process may also aid to explain why no eggs or krill larvae were observed in the sediment trap used in this study (L165-166).

L180-181 Please explain how this estimation was determined either in the methods or briefly here.

METHODS

L203: Please include the preservatives used in the supernatant (may fit best in first paragraph of methods).

L204-206: Change sentences to, "Prior to splitting, "swimmers" (i.e., zooplankton that can enter the receiving cups while alive) were carefully removed. Samples were first ...

L219: Should be i.e.,

L241: Undo capitalization of "The".

L250: Change "through" to "by".

L256: Should be i.e.,

L267: Change "through" to "by".

L269: Period after 2018).

FIGURE CAPTIONS

L513: Please state in the caption for Figure 3 that only select months are shown. Is there a reason not all months were included?

L515-516: Change caption to, "Monthly mean biomass determined from an inverse model calculation based on exuviae caught in the sediment trap."

FIGURES

Figure 1. This may be a formatting issue converting the figures into a pdf but some of the bars in the error bars and barplots are cutoff. If these figure conversions are correct, please adjust the y-axes of figure 1 subplots to include the error bar at the peak of the POC flux (1a), the bar outline for months Oct. and Nov. (1b), and the top of the error bar for winter (1c). Also, in the Figure 1 figure caption it states standard deviations are shown for 1b & 1c but they are not included in the figure as they are in 1c. Finally, the legend text for carcasses in Figure 1b should be capitalized to be consistent with the capitalization used for FP and Exuv.

Figure 2. While ascetically pleasing, I find the information conveyed in this conceptual diagram could more easily be stated in a table. If the authors added more complexity to the diagram (i.e., how the primary krill carbon vectors varied by season) then I think it would be justified to keep.

SUPPLEMENTARY DATA

Note: The title for both supplementary files are different than in the primary manuscript text.

REVIEWER COMMENTS

Reviewer #1 (Remarks to the Author):

The manuscript entitled “Continuous moulting by Antarctic krill drives major pulses of carbon export in the Scotia Sea, Southern Ocean” uses sediment trap samples to assess the contribution of the flux of krill exuviae, fecal pellets, and carcasses to total particular organic carbon (POC) flux. It sheds important light on the role of Antarctic krill in mediating carbon transport and sequestration and shows that carbon flux through sinking krill exuviae is nearly equal to that of their sinking fecal pellets. This is an important finding as it increases our understanding of the Southern Ocean as a carbon sink.

I think that the use here of sediment trap samples to demonstrate the importance of exuviae and krill moulting to carbon flux is strong. I have some reservations, however, with the approach taken to estimate krill biomass and abundance from the exuviae found in the sediment trap cups (it could just be how the approach has been explained). I provide more details on this below. I have additional concerns over other aspects of the methods too that I describe below. On the whole, I think the manuscript is well written, novel, and relevant to the scientific community. I recommend that the manuscript be reconsidered after major revision.

AU- We thanks the reviewer for all the constructive comments and to highlight that the methodology was not fluent enough. In the specific, concerning the equations and step to calculate Krill biomass from exuviae collected into sediment traps we agree that it did include some jumps in logic that were not clearly explained. We have now completely revised this section (also on the base of another reviewer suggestion) and we also include much greater detail on this calculation in an extensive Supplementary Materials document SM2. The SM2 document include extensive information on: i-Determination of intermoult period, ii-The equivalent wet weight (WW) of krill, iii-Spatial scale of capture by the sediment trap, iv-Mean estimate Antarctic krill biomass, v-Deriving reasonable upper and lower bounds around the mean estimate. Please see more detailed responses below.

Specific Comments:

R-Lines 32-38: This paragraph doesn't flow well, and I would recommend revising it for improved readability.

AU-We revised the paragraph as follow: Sinking carcasses of krill can further contribute to the oceanic carbon export (especially outside the phytoplankton growth seasons), becoming a major food source for the benthos (Tang et al. 2014). Carcasses result from predation and/or non-predatory mortality such as senescence and starvation as well as the presence of a number of both external and internal parasites (Mauchline and Fisher 1969, Gomez-Gutierrez et al. 2003).

R Line 33: What do you mean by “predator activity”? Is this just predation?

AU We changed predator activity with predation

R Line 56: Remove “krill” after “circumpolar” so that it doesn't read “krill stock of ...krill”.

Line 81: I think “varied” is better here than “altered”.

AU Changed as suggested

R-Line 85: What months did you use to get averages of each season?

AU- The first bottle open the 15 Jan. We then average each season as follow: bottle 1-2-3-14-15-16 (Summer, from Dec to Feb), bottle 4-5-6 (Autumn, from March to May), bottle 7-8-9 (Winter, from June to Sept), bottle 10-11-12-13 (Spring, from Sept to Nov). The reviewer is right this information was missing in the manuscript. To clarify in the text i-We added how season were allocated according to the sample bottles in the methods; ii-We added the bottle opening rotation time in the supplementary material in the table 1.1.

R-Line 91: Did you use a statistical approach to test whether the range in lengths was similar throughout the study period or was this just eyeballed? It seems it was just eyeballed, but it would be better to statistically test this.

AU-We have inserted some more detailed text in this regard, identifying the months where significantly different standard lengths were apparent and detailing the statistical tests carried out: “Estimated krill standard lengths were between a minimum of 15 mm and a maximum of 44 mm over the study period, with a median body length across all months of 30 mm (Fig. 3). No significant differences were found in the distribution of standard lengths between months with the exception of March, where median standard length was 24 mm and October, where it was 41mm, although exuviae numbers collected the latter month were relatively low (Kruskall-Wallis 1-way ANOVA on ranks, $H = 102.78$, 14 df, $P < 0.001$, Dunn’s Method post-hoc test, March vs October, $Q = 4.066$, $P = 0.005$)”

R Line 93: It is certainly possible to have mature adults that are within this size range though.

AU Agreed and we have modified the text accordingly “Standard lengths between 15 and 44 mm are typical of juvenile and sub-adult krill in this region, although smaller mature adults can also fall within this size range (Tarling et al. 2016)”

R Lines 99-100: I think you mean the other way around here – “maximum and minimum levels of biomass occurring in the summer and winter, respectively”.

AU Yes, this was the other way around. We have corrected the mistake.

R-Lines 100-101: I’m not sure I agree with this approach. It is possible that the exuviae you observed in the winter came from the same biomass of krill estimated from the summer. I don’t think summing the biomass across seasons is a good approach.

AU-We have now deleted reference to an integrated annual biomass estimate and now only refer to the 15 to 30 day estimates over the annual cycle

R Line 103: Add a period/full stop at the end of this sentence.

AU Done

R Line 110: Needs rephrasing – I suggest addition of words in bold “which is an order of magnitude greater than that observed even in highly productive...”.

AU Change done has suggested

R Line 122: Why “must” this “outlast the proceeding winter”? And can you show that with your data? Do you rather just mean that it is likely sufficient to fuel the benthos through the winter? And if so, what references can you provide to support this?

AU We cannot support the fact that this carbon input will be sufficient to fuel the benthos during the winter. We clarified the meaning of the sentence as follow “may be an important source of nutrition to fuel the benthos through the winter”.

R- Lines 127-128: Doesn't the krill fishery start in the summer? Can you say that it extends into the winter, therefore large aggregations must, in most years, persist into the winter in the region?

AU- The krill fishery around South Georgia only operates during winter when the spread of sea ice restricts access to the preferred fishing grounds further south (Kawaguchi *et al.*, 2009). Closure of the fishery during summer has now been adopted as a management measure with the Government of South Georgia and the South Sandwich Islands (GSGSSI) marine-protected area management plan to avoid competition with seabirds and seals during the breeding season (Collins, 2012). Winter fishing is permitted on the rationale that dispersal migration of krill-dependent predators away from their colonies during this period will greatly reduce the potential for competition, although this assumption has not been tested empirically.

Collins, M.A. (2012) South Georgia and South Sandwich Islands marine protected area management plan. Government of South Georgia and South Sandwich Islands, Stanley, Falklands;
Kawaguchi, S., Nicol, S., Taki, K. & Naganobu, M. (2006) Fishing ground selection in the Antarctic krill fishery: trends in patterns across years, seasons and nations. *CCAMLR Science*, 13, 117– 141.

R Line 133: The citation here seems odd having “- Siegel book chapter” tagged onto it. If this is the way Nature Comms requires book chapter citations, then disregard this comment, but if not, then correct the citation please.

AU Siegel book chapter was deleted

R Line 137: Either “lead to a reduction in” or “lead to reductions in”, but not as currently written.

AU We changed as suggested

R: Line 149: I believe the work of Reiss and colleagues demonstrates the move of krill biomass inshore into the Bransfield Strait region in winter. Order of magnitude higher biomass in this region in winter compared to summer.

AU: We thank the reviewer for suggesting the work of Reiss et al. 2017. We added the following sentence in the discussion “Most recently Reiss et al. (2017) observed an order of magnitude increase in krill biomass in the Bransfield Strait in winter suggesting that this increase must have resulted from active horizontal migration from offshore feeding and spawning areas occupied during summer”,

Reiss CS, Cossio A, Santora JA, Dietrich KS and others (2017) Overwinter habitat selection by Antarctic krill under varying sea-ice conditions: implications for top predators and fishery management. *Mar Ecol Prog Ser* 568:1-16

R Lines 150-152: A little awkwardly phrased. I suggest “We suggest that the large decrease in exuviae and FP flux in the winter is likely due to a combination of the biological and physical processes mentioned above.”

AU Changed as suggested

R-Lines 165-166: Is this surprising given your sediment trap was at 300m and female krill typically only release eggs over waters that are >1000m?

AU- Although offshore spawning migrations have been reported for the Antarctic Peninsula, this is not necessarily the case elsewhere. Since embryo and larval development rates depend on

temperature (Ross et al, 1988), it is likely that krill larvae can hatch at shallower depths in the warmer waters around South Georgia and hence we think that it is worth reporting whether or not krill eggs have been collected by the sediment trap in the present study region.

Ross, RM, LB Quetin, E Kirsch (1988) Effect of temperature on developmental times and survival of early larval stages of *Euphausia superba* Dana. J. Exp. Mar. Biol. Ecol., 121: 55-71, doi: 10.1016/0022-0981(88)90023-8.

R-Line 168: The way this is phrased gives the impression that the juveniles developed from eggs spawned that same season, which would not have been the case as they first must survive a winter. Do you rather mean that these were larval krill? The size range suggests at least some of them may have been. And that would make more sense.

AU- A krill egg hatches and develops through several larval stages before becoming a juvenile krill (~15 mm, Siegel, 1987). Recruitment at South Georgia is not the same as at the Peninsula; instead of local recruitment, krill is delivered to South Georgia from the Antarctic Peninsula and upstream islands (e.g. Murphy et al. 2007). However, we realised that the way we use the word “recruitment” here was a bit confusing. To clarify this point we added this sentence into the main text: “Neither eggs nor larval stages of krill were found in our sediment trap samples at any time of year indicating an absence of successful spawning in this region. We did observe a decrease in the size of exuviae within the sediment traps during autumn (March), indicating a recruitment of juvenile krill into the study region. These krill were juvenile (15+ mm) and are most likely to have originated from areas upstream of South Georgia, such as the Antarctic Peninsula and outlying islands (Murphy et al., 2007a, Fielding et al. 2014, Reid et al., 2010).” Siegel V (1987) Age and growth of Antarctic Euphausiacea (Crustacea) under natural conditions. Marine Biology 96, 483-495.

R-Lines 168-171: I’m not convinced that the evidence provided here supports allochthonous recruitment.

AU-We agree that the absence of eggs in the sediment trap is not direct support for the working hypothesis that krill at South Georgia rely on allochthonous recruitment. However, the fact that we did not find any eggs means that there was no evidence to contradict this working hypothesis. Nevertheless, we have taken the term “allochthonous recruitment” out of the revised paragraph and report only the facts that successful spawning did not take place in the region and that the abundance of juveniles increased in March, with the most likely source being upstream of South Georgia.

R-Line 171: Typically, the new recruits appear in Spring after overwintering as larvae not in autumn on the same year they were spawned.

AU- We accept that using the term recruitment may be misleading in this context and, with respect to the aims of the present manuscript, is unimportant. We have removed this term and simply report that juvenile abundance increased in March, which coincided with a secondary peak in FP flux.

R-Line 192: Which months were classified as summer and which were winter?

AU- Please see earlier response to line 85. Information about how seasons were allocated have now been included in the method section.

R Line 196: Should be “these data” rather than “this data” since data is the plural of datum.

AU We changed the typo.

R-Line 233: “Krill total length” should rather be “Exuviae total length” shouldn’t it? Since the length has been estimated from the exuviae uropod rather than a live krill uropod. It is also necessary to include the caveats of doing this – i.e. that you aren’t actually getting length measurements of live krill, but rather of their exuviae.

AU-The reviewer is correct in that we are estimating the standard length of the exuviae which is not necessarily the same as the standard length of the live krill since moulting. Nevertheless, given that growth in length is comparatively small at each moult and may, in some instances, even be negative (Tarling et al. 2006, Atkinson et al. 2006), we considered it a fair assumption for the purposes of the present calculation that exuviae length was approximately equal to the length of the live krill that moulted. We have now inserted this explanation into the text.

R Line 236: I believe it should be SL for Standard Length, rather than S1.

Lines 238-240: As for comment above – SL not S1.

AU We have retained S1, since this was the nomenclature used by Miller (1983) and refers to an internationally accepted standard measurement of krill length.

R Line 239: “from exuviae (v)” – v is not in the equation but is provided as if it is.

AU We have rescripted many of the equations based on feedback from the present and other reviewers – the term v is no longer relevant in this context.

R Line 240. Sentence is missing a period/full stop.

AU Added.

R-Lines 252-253: Why would it be assumed that they moult randomly to each other? If there is a citation for this, please provide it here. And, by stating this, are you also saying that only one moults on a given day? If so, this is definitely not the case.

AU-Random moulting means that the moulting day of krill α is independent of all other conspecifics in the population. It implies that there is no interdependence between individuals on when each enters its moulting phase (ecdysis) and releases the exuvia. Rather, the rate at which each individual goes through its moult cycle, and enters ecdysis, is dependent on factors such as its size, sex and temperature experience, which will vary between individuals within a population and results in a population pattern of moulting that cannot be distinguished from random. This has been shown to be true by histological studies of Buchholz et al. (1989) and observational studies by Tarling et al (2006). We now cite both studies in relation to this aspect.

R-Lines 253-256: IMP is the number of days between moults for an individual krill, and $1/IMP$ is the number of moults in a given day for that same individual (and would be a fraction of 1 unless they’re moulting more than once a day). I don’t think I’ve ever seen $1/IMP$ being referred to as the number of krill that moult on any given day. I don’t see where this jump in logic comes from. How can the inverse of the inter moult period, the number of days between moults, become the number of krill that moult in one day? Perhaps I’m missing something here, but that would suggest that this section needs to be revised for clarity.

AU-We have now completely revised this section given that it did include some jumps in logic that were not clearly explained. We also include much greater detail on this calculation in an extensive Supplementary Materials document SM2.

R-Lines 267-269: I'm not sure why this is mentioned here as you don't present the data in this format.

AU-The description of the calculation method has been altered in the revision and this comment is no longer relevant.

R Line 269: Period/full stop missing from the end of the sentence.

AU Added.

R-Line 272: Please clarify how seasons were allocated somewhere in this manuscript.

AU- We have already replied to this comment. Information about how seasons were allocated has been included in the method section.

R-Line 273: Just a quick look at the data for POC, exuviae, and FP especially (maybe not as much for carcasses), suggests that the data are not normally distributed. Also, I used the data you provided in the Supplemental Materials and looked at normality myself, and the data are very clearly skewed. Can you provide the p-values for these tests for normality and equal variance that were passed please? Or maybe they were transformed already?

AU-Data are not normally distributed, having NOT passed Prior tests for normality and equal variance. There was a typo in the text and we apologise for it. We decided to choose the nonparametric version of the test which does not require the assumption of normality. The text has been changed accordingly: Kruskal-Wallis H Test was used to determine whether there were any significant differences between seasons with regard to POC, FP, carcasses and exuviae. Differences were considered significant where $p < 0.05$. H and p-values showing that differences are significant have been added into the text

R Line 508: Period/full stop missing before "c)".

Line 509: Period/full stop missing at the end of the sentence.

Line 520: Period/full stop missing at the end of the sentence.

Line 521: Period/full stop missing at the end of the sentence.

AU all the missing full stops have been added.

R-Figure 2: Looks quite strange with the error bars overlapping like that – can you have grouped bars rather than stacked?

AU- We think the reviewer is referring to Fig 1c here. We prefer to keep Fig 1c like it is because grouping the bars makes it harder to tell the difference between the total of each group.

R Supplementary Data – why are their asterisks in the units? E.g. $\text{mg} \cdot \text{m}^{-2} \cdot \text{d}^{-1}$ should just be $\text{mg} \cdot \text{m}^{-2} \cdot \text{d}^{-1}$

AU Changed as suggested.

R Supplementary Information – I think this map figure would be more useful in the manuscript itself.

AU We appreciated the suggestion of the reviewer but we prefer to leave the map in the supplementary information.

More detailed comments/questions on methods:

R-1. Were the trap samples preserved in formalin in situ? If not, how did you account for bacterial degradation? If they were preserved in formalin in situ, is it not possible that a krill could swim into

the cup and become preserved, rendering it a “carcass”? There is no mention of preservation, but I know from other work that this is typically done. Please provide the details needed.

AU-1-The trap bottles have been filled with buffered formalin before deployment for in situ preservation of sinking material (as standard procedure for sediment trap deployment). We added the following sentence in the methods for clarification “Each bottle contained a solution of 4% formalin with 5g Sodium Tetraborate (BORAX) mixed in to arrest biological degradation during sample collection and to avoid carbonate dissolution”. Active intruders that swim into traps and die in poisoned traps are appropriately not included in the flux and should be removed. As we have explained in the main text “Prior to splitting, “swimmers” (i.e., zooplankton that can enter the receiving cups while alive) were carefully removed: samples were first wet-sieved through a 1 mm nylon mesh and the remaining swimmers hand-picked under a dissecting microscope”. To discriminate between krill “carcasses” and “swimmers”, we only identified as carcasses the individuals showing clear signs of decay and/or damage (e.g. broken antenna, legs, etc.).

R-2. What were the temperatures for each time period that were used in the calculations of IMP? Please provide these in your manuscript or supplemental materials.

AU-2 Now included in a table in Supplementary Materials SM2 Table S2 and S3.

R-3. How were the “winter” and “summer” months allocated for the 15 and 30-day time periods? E.g. was April considered winter?

AU-3 As for previous response to reviewer comment April was considered autumn and information about how seasons were allocated has been included in the method section.

R-4. Supplementary Data 1.3 – My understanding from the Methods section is that one biomass was calculated for each time period. If that is true, how could there be an SD associated with the value? If this is not true, and biomass shown here is an average for multiple days of the time period, how was this calculated? If you took sub-samples from each cup, that would be considered pseudo-replication.

AU-4 The data 1.3 represent the biomass calculated for each bottle. The SD is associated with the sub-samples taken from each trap bottle. The SD shows that the samples from each bottle were homogeneously split in order to proceed with further analysis. This is a standard procedure for sediment trap samples. To clarify we added the word pseudo-replicates in the method session as follow “Each sample was then divided into a series of replicate fractions (pseudo-replicates)”. Also we add the following sentence in the caption to Fig 1a: Error bars are standard deviations from replicates sub-samples from each cup.

R-5. I tested out the equations given in the Methods section to estimate krill biomass for April given the available uropod lengths provided in the supplementary materials. I assumed since only 4 were measured, that there were only 4 moults in the trap cup for April. I had to guess at the temperature (please provide those for each month) and had to assume that April was part of “winter” and therefore the time period was 30 days. I applied the equation provided to convert uropod lengths to standard length, and then averaged the standard lengths (30.75mm). I then use Tarling’s equations to calculate the moult probability for juveniles, males, immature females, and mature females for that average SL and a guessed temperature of 1°C, and then calculated the average moult probability from that – it was 0.33. Then, I calculated the average IMP for April as $30/0.29=91.10$ days. I calculated the value for V (integrated krill) as $4/0.5m^2/30$ days to get 0.267 ind./m²/day. With this, krill abundances, K, were calculated as the average IMP-1 x V to get ~24 ind./m². I applied the equation to calculate biomass, W, and reached a biomass of 0.001 g/m². When I compared this

to the value provided in the Supplementary Materials for April, it was very different (3.52 g/m² is provided there). I also tried it with different temperatures and with the period set to 15 days, in case I'd gone wrong there. I'm now very confused. Either the methods need to be more clearly explained, or there is an error somewhere.

AU-5 We are grateful for the reviewer's thoroughness in testing out the equations to verify that we have reported them correctly. In the following, we will work through the calculation for the same month. There were 4 exuviae caught in this period, with uropods measuring 5.02, 4.53, 4.05 and 3.64 mm. According to Eq 1, this equates to 34.5, 31.9, 29.4 and 27.2 mm standard length (S1). As we now report in Tables S2.2 and S2.3, the seawater temperature for that period was 3.2°C. According to the equations of Tarling et al. (2006), IMP can be calculated when S1, temperature and sex/stage is known. Given that we are unable to determine the latter, we take the average across all possible stages. From Tarling et al (2006, their Table 3) we get a sex/stage-averaged probability moulting over 5 days (P) of 0.35, 0.36, 0.38 and 0.39. As clearly stated in Tarling et al. (2006, their Table 3), P must be divided into 5 to obtain an IMP, which becomes 14.8, 14.3, 14.0 and 13.8 days respectively (we note here that the reviewer divided P into 30 to obtain values of ~90 days). As stated, there were 4 exuviae captured over a 30 day period, giving an average of 0.13 exuviae per day. Assuming an even distribution over the spatial scale of sampling, we divide this by 0.5 m² to give 0.27 exuviae m⁻² d⁻¹. Multiplying this by the IMP gives a krill concentration of 3.799 individual m⁻². However, in this particular example, we also factor for winter quiescence in moulting, following Buchholz et al. (1989), which increases IMP by a factor of 5, hence making the value 3.799 x 5 = 18.995 ind. m⁻². For the respective standard lengths for the 4 specimens, the average wet weight was 0.187 g. Multiplying this against the concentration of krill gives 3.5 g m⁻². We have endeavoured to document these equations as faithfully as possible. We have made a note in the Supplementary Materials SM2 that d = 5 in Table et al (2006, their Table 3) to avoid any confusion. However, we will not reproduce Tarling et al. (2006, their Table 3) in SM2 since the publication lays it out clearly and is open access.

Reviewed by Professor Kim Bernard, Oregon State University

Reviewer #2 (Remarks to the Author):

The manuscript by Manno et al. find support for previous observations that krill fecal pellets can be a major contributor to carbon export in the Southern Ocean. The novelty of the manuscript is that in addition to krill fecal pellets, krill exuviae from continuous moulting by krill may be an important source of carbon export in the Scotia Sea. This is a mechanism that has so far been overlooked in the material collected in sediment traps. The carbon export by krill exuviae can match the carbon export from krill fecal pellets as observed by Cavan et al. (2015) and Belcher et al. (2019), making krill the dominant source for carbon export in the Scotia Sea. The manuscript presents an overlooked potential pathway for carbon export in the Southern Ocean and suggests that data sets from long-term moored sediment traps may offer a method to evaluate the impact from krill exuviae for carbon export in future studies, and potentially from past collected samples, if the material is still available.

AU We are very grateful to the reviewer for their comprehensive suggestions on how to improve the manuscript. In the specific, we include much greater details on the methodology which describe the calculation of krill biomass from exuviae collected in the sediment traps. Extensive detail about our methodological approach is now provided in a Supplementary Materials document SM2. The SM2 document include information on: i-Determination of intermoult period, ii-The equivalent wet

weight (WW) of krill, iii-Spatial scale of capture by the sediment trap, iv-Mean estimate Antarctic krill biomass, v-Deriving reasonable upper and lower bounds around the mean estimate. Please see more detailed responses below.

Specific Comments:

R- The author measured total POC flux from the sediment trap samples and collected exuviae for POC measurements. Unfortunately the authors used conversion factors to evaluate the role of both krill fecal pellets and carcasses to estimate their contribution to total POC export. It would have been a much stronger data set if the authors had also measured the POC of the fecal pellets and carcasses. Especially the fecal pellet carbon would have improved the manuscript, since those are most likely to change seasonally depending on the food source from which the fecal pellet were produced.

AU- We realised this approach is not well described in the text and there is some missing information which leads to the reviewer's concern/criticism. We take the opportunity here to clarify our approach. Although we did not directly measure the POC content of the krill FPs in this study, the conversion factors that we used to derive carbon content (from Manno et al., 2015) were generated using an elemental (CHN) analyser to estimate the carbon content of euphausiid FPs from sediment trap samples deployed at oceanic stations upstream and downstream of South Georgia. On the basis of the CHN results, Manno et al (2015) determined seasonal conversion factors for two periods: late spring to early autumn (October to April, $0.030 \text{ mg C mm}^{-3}$) and late autumn to the end of winter (May to September, $0.018 \text{ mg C mm}^{-3}$) reflecting the difference in C content due to changes in the food sources during the seasons. For this reason, the fact that i) the conversion factors come from direct analyses of C content of FPs collected from sediment traps in the Scotia Sea, ii) the conversion factor is specific for mainly Euphausiacea and iii) the conversion factor discriminates between high productivity/food and low productivity/food periods, we are confident that our indirect calculation of C is a good estimation of the C content of the krill FPs from our samples. We have added a summary of this information to the method section to clarify these points.

R-The estimations for the krill standing stock is very hard to follow. The authors refer to previously published equations by Tarling et al. (2006), however, Tarling et al. (2006) is not cited in the reference list, as far as I could determine. I found the Tarling et al. (2006) paper - assuming it to be: "Natural growths rate in Antarctic krill (*Euphausia superba*): I. Improving methodology and predicting intermolt period" published in *Limnology and Oceanography*: 51(2): 959-972. Only after going carefully through the equations in the current manuscript and the equations in Tarling et al. (2006), it became somewhat clear how the authors had estimated the krill standing stocks. Here there are several assumptions made by the authors of Manno et al. (current manuscript) in order to get the standing stocks. It is not clear to me how the authors addressed one of the major premises presented by Tarling et al. (2006) in order to use the method, where the number of moltings per day must be constant and equal to the inverse of the molt duration. How did the authors ensure this criteria?

AU-The citation of Tarling et al (2006) has now been included.

We have also dealt with a similar query from another reviewer (above) and refer to the arguments made there. Nevertheless, we extend this argument here to tackle the specific query being made. The assumption that the number of moultings per day being constant and equal is a tacit assumption of all studies that have estimated the intermolt period of euphausiids (Quetin and Ross 1991, Nicol et al. 1992, Kawaguchi et al. 2006). This was tested within Tarling et al (2006) which found the assumption to be broadly true within their experimental population. Furthermore, in a histological

analysis where the distribution of a krill population over the moult cycle was analysed, Buchholz et al. (1989) did not find there to be a concentration of krill in any one phase of the cycle, ruling out any moult synchronicity. There is some anecdotal evidence of mass moulting, where there is a synchronous release of moults from a startled swarm (Hamner et al. 1983), but this does not imply that all krill within the swarm moulted. Krill can only moult if they are in the late premoult stage of the moult cycle, which lasts around 1 day (Cuzin-Roudy and Buchholz 1999). For an average IMP of 14 days, the reported mass moulting likely only involved 1/14th of the total numbers of krill in the swarm, which is still a substantial number of moults to an observer and an effective decoy (as they infer). Therefore, there are a number of independent studies supporting the assumption that the day of moulting of one krill is independent of all others in a population and presently no evidence to refute this assumption.

R-Another aspect which was not clear to me was how the authors estimated the capture area of the trap? To me it seemed that the authors assumed that all exuviae that were captured by the trap sank vertically without any horizontal movement? Typically a sediment trap collect material from an area larger than its funnel opening, this would make their estimated standing stock considerably smaller than the one estimated in the current manuscript? This is especially related to the method description from line 241 to line 269. Here the authors first calculate the total concentration of krill per square meter above the trap - this seem that the authors assume all krill to be distributed in a water column of one square meter above the trap, surely this area has to increase as you are further away from the sediment trap and closer to sea surface.

AU-We concede that the description of this calculation required substantial improvement and we have completely rewritten this section and also now include a substantial Supplementary Materials document (SM2) detailing all steps in this calculation. To answer the specific query, we do not assume that the krill directly above the sediment trap are those which contributed to the exuviae caught by the trap. We entirely agree that the passive horizontal advection of these exuviae as they sink through the water column will mean that they will have originated some horizontal distance away, which we have modelled to be around 1 to 2 km (see SM2). However, assuming that, on a 1 to 2km spatial scale, the distribution of krill is approximately even when integrated over a 15 or 30 day period, then one can assume that the concentration of krill within the 0.5 m² capture zone of the sediment trap is the same as elsewhere at that spatial scale. This is fundamentally the same assumption made by all sediment trap studies. Nevertheless, we also now include calculations where the assumption of even distribution is violated and that there is an over-representation or under-representation of the true amount of krill within the spatial scale of capture based on the amount of captured exuvia (see below for further details on this calculation).

R-The next aspect is when the authors calculate the concentration per cubic meter in the upper 50 m of the water column. Here the authors argue that all krill are within the top 50 m of the upper water column, which again may be seasonally dependent, but is a reasonable assumption. Here I would like to see what the "collection funnel" in the water column above the trap was, dependent on the current velocities, and then see the authors provide a range for the krill standing stock and biomass in the water above the trap.

AU-As mentioned in the response above, SM2 now includes the description of a model that calculates the horizontal distance travelled by the exuvia during their descent to the depth of the sediment trap. The model is based on empirical observations made by an acoustic Doppler current profiler moored above the sediment trap for the duration of the present study. The model calculates the horizontal distance travelled by a typical exuvia on each deployment day, which is mainly between 1 and 2 km (outer limits of between 70 m and 4 km) which we define as the spatial scale of

capture. Within this scale, all moulted exuviae have the potential to be captured by the sediment trap. Not all potential captures will be realised however, since a number of forces act on the sinking exuviae, moving them in and out of the “capture zone”, i.e. the zone which figuratively lies within the mouth area of the sediment trap (Fig. S2.1). How representative the number of captured exuviae are will depend on the distribution of krill within the spatial scale of capture (i.e. whether patchily or evenly distributed) and the time period of capture over which the results are integrated. We firstly calculate a mean biomass estimate based on the assumption that various physical and biological factors will act to modulate any unevenness in distribution over the 15 to 30 day period of sample integration. Nevertheless, we now also calculate outer bounds to this estimate through adopting scales of unevenness from the South Georgia acoustic surveys of Fielding et al. (2014) which analysed spatial cells of 500 m length and determined the coefficient of variation between them. Fielding et al. (2014) including 16 such surveys in their study, calculating a CV from each. We extracted the 95% percentile CV value from their results and inflated or deflated the daily number of captured exuvia in the present study accordingly to determine upper and lower bounds to the mean biomass estimate.

R-Overall, the authors present an interesting new mechanism, but it is to a large extent based on few actual measurements and many conversions and assumptions. The authors should address the assumptions and discuss their impact on their findings.

AU-We accept that a number of conversions and assumptions are required to arrive at a biomass estimate. However, we provide supporting evidence for each of these steps and now provide further analyses to provide upper and lower bounds based on empirically measured scales of unevenness in krill distribution for this region.

R The manuscript is well written and I have only found a few issues to the text itself:

Line 32-34 could use a citation to the statement.

AU We re-organize the text as follow with the appropriate references: Sinking carcasses of krill can further contribute to the oceanic carbon export (especially outside the phytoplankton growth seasons), becoming a major food source for the benthos (Tang et al. 2014). Carcasses can be due to predation and/or non-predatory mortality such as senescence and starvation as well as the presence of a number of both external and internal parasites (Mauchline and Fisher 1969, Gomez-Gutierrez et al. 2003).

R -Line 38: I guess Tang et al. (2014) was studying copepods and not krill - are there any studies of krill?

AU-The observed behaviour of krill to move to the deep seabed have been highlighted as an important source of nutrients and carbon for the deep-living pelagic and benthic fauna in deep sea environment. Hirai J, Jones D (2012) The temporal and spatial distribution of krill (*Meganyctiphanes norvegica*) at the deep seabed of the Faroe-Shetland Channel, UK: a potential mechanism for rapid carbon flux to deep sea communities. Mar Biol Res. However to our knowledge there are not any studies that specifically focus on the relevance of carcasses of krill sinking in supporting the benthos community. For this reason we believe that the use of the work of Tang et al. was appropriate to highlight as krill carcasses (as for the case of copepods) can be a relevant source of food for the benthos due to their larger size and sinking rate compared with copepods.

R- The estimations for krill standing stock should be explained more detailed so the reader does not have to go carefully through Tarling et al. (2006) in order to understand what was done in the

present manuscript. Assumptions should be better highlighted and discussed (see above).

AU-We have fully addressed this point in our responses to the previous comments (see above).

R -Tarling et al. (2006) needs to be included in the reference list.

AU-We added The reference "Tarling, G. A. ; Shreeve, R.S.; Hirst, A. G.; Atkinson, A.; Pond, D. W.; Murphy, E.J. ; Watkins, Jon L.. Natural growth rates in Antarctic krill (*Euphausia superba*): I. Improving methodology and predicting intermolt period. *Limnology and Oceanography*, 51 (2). 959-972 (2006)"

R- It is surprising how much of the seasonal flux (even in early spring) was contributed by krill (either as fecal pellets or exuvia). What was the rest of the exported material? Can you perform POC measurements for krill fecal pellets at different seasons?

AU-The rest of exported material consisted mainly of phytoplankton detritus, radiolarian, empty pteropods shells, foraminifera and other FP (most of them with oval shape). So there was a fraction accounting for Biosilica flux and carbonate flux as well which we do not present in this study because it is out of the scope of our investigation. The huge contribution of krill to the flux suggests (especially in the spring) the presence of a very efficient export mechanism where a large amount of the available phytoplankton and small zooplankton is ingested by krill swarms and egested as FPs.

Note: we have already replied to the comment related to the measurement of POC in FPs, and the way in which we take into account the seasonal change in FP carbon content has now been described in detail in the methodology.

R- At times in the manuscript it is unclear if the authors are referring to dry weight or wet weight.

AU-We refer to wet weight when we talk about krill biomass. This has been clarified in the different sections of the main text

R- I miss a paragraph in the discussion addressing potential climate change and its role on the observed mechanism? What happens when the Southern Ocean is warming and the habitat of krill will be restricted by warmer water temperatures? Can you learn anything from your seasonal data?

AU-We agree with the reviewer that a paragraph addressing the potential impact of climate change will be a nice addition and we added the following text:

Increasing water temperature in the Scotia Sea, as a result of climate Cchange, will likely have a negative impact on krill growth and biomass (Hill et al. 2013, Kelin et al. 2018). Here, we show for the first time the crucial role of krill exuviae as a vector for C flux in the Southern Ocean, a region which contributes significantly to the global C export production (Schlitzer et al. 2002). Thus a potential decrease in Kkrill biomass is likely to impact the marine biogeochemical cycles. Further, since the krill moult cycle (and in turn exuviae production) depends on temperature (Buchholz et al. 1991), our findings highlight the sensitivity of C flux to rapid regional environmental change (Schiermeier, 2010; Flores et al., 2012).

New references:

-Hill SL, Phillips T, Atkinson A. Potential climate change effects on the habitat of antarctic krill in the weddell quadrant of the southern ocean. *PLOS ONE*. 2013;8(8): e72246. pmid:23991072

-Klein ES, Hill SL, Hinke JT, Phillips T, Watters GM (2018) Impacts of rising sea temperature on krill increase risks for predators in the Scotia Sea. *PLoS ONE* 13(1): e0191011.

<https://doi.org/10.1371/journal.pone.0191011>

-Schlitzer, R. Carbon export fluxes in the Southern Ocean: results from inverse modeling and comparison with satellite-based estimates. *Deep Sea Res. Part II Top. Stud. Oceanogr.* 49, 1623–1644 (2002).

R -I believe that the authors are having a long time-series of sediment trap data, why only present one year for this mechanism? The authors could have investigated several years, at least for the summer period when krill exuviae seem to play a crucial role for the total exported POC?

AU-The reviewer is mistaken. This was the only year a sediment trap was deployed on this mooring platform. The reviewer is probably referring to the long-time series sediment traps (named P2 and P3) deployed at 2000 m which are located in the oceanic sector of Scotia Sea while the mooring platform of this study is located on the Shelf region of South Georgia (sea floor depth of 400 m). We recognise the relevance of investigating several years, however it is not always logistically and economically possible to support long-term mooring platforms (and this is especially true in remote regions such as the Southern Ocean). However, the one-year sediment trap data still represent a valuable seasonal and innovative dataset. The results of this study are promising and support the importance for developing future long term study in this region in order to fill the gap in the seasonal study of krill populations on interannual scale.

Reviewer #3 (Remarks to the Author):

General comments:

This manuscript represents a well-written, useful study analyzing the contribution of Antarctic Krill, *Euphausia superba*, contribution to particulate organic carbon export (POC) in the Scotia Sea of the Southern Ocean. The major findings of the study show that krill exuviae flux is of similar orders of magnitude to that of fecal pellet flux, which indicates previous studies only quantifying krill fecal pellet flux may underestimate the overall contribution of krill to POC flux. The novel aspect is that no study to date has quantified the contribution of krill exuviae to POC flux in the Southern Ocean. In addition, the study utilizes an inverse model to estimate krill population size based on measured exuviae collected in the sediment trap. These data presented in this study will be valuable to researchers interested in Antarctic krill zooplankton, and to those interested in quantifying zooplankton contributions to the biological pump in the Southern Ocean. This study will influence thinking by encouraging other research programs with sediment traps in the Southern Ocean to quantify fecal pellets and exuviae when determining krill contributions to POC flux. While the claims are convincing, some of the broader conclusions are oversold and need to be better qualified or removed. The sensitivity of POC flux to environmental change is not supported by the dataset in this study and the connection that krill play in contributing to atmospheric carbon drawdown needs to be better explained. Overall, the study is sound and answers the call by Cavan et al. 2019, *Nat. Commun.* to better parameterize Antarctic Krill contributions to biogeochemical cycles.

AU-We thanks the reviewer for the positive feedback and all the comments and suggestion. We carefully address the concern of the reviewer about the conclusion and we re-organize the text accordingly (see detailed responses below).

Specific comments:

ABSTRACT

R L7: Use more active language in abstract... “Antarctic krill are important in...” or “Antarctic krill play an important role in...”

Au Changed from Antarctic krill have to Antarctic krill play.

R L11: Include specific location...in the Scotia Sea of the Southern Ocean.

AU Location added as suggested.

R L15-17: This conclusion is vague. Also, the study does not analyze the effects of environmental conditions on moulting rate so the way the sentence is currently stated is unjustified.

AU- Despite the present study not analysing the impact of environmental conditions (such as change in temperature) on moulting rates, our results show the important role of exuviae in promoting the transport of carbon. In fact, we demonstrate that krill exuviae can contribute to the carbon flux with an amount equal to FPs. Since it is already known that exuviae production depends on temperature (Buchholz et al. 1991), the exuviae carbon component can be strongly influenced by environmental change (e.g. increase in temperature). This in turn can impact the POC flux. Thus we do not believe our conclusion to be vague but that instead it suggests a potential wider implication of our results. Rather than claiming our results “support”, we carefully worded our text to say that our results “highlight” the sensitivity of POC to environmental change. However, we appreciate that the sentences related to the conclusion in the text (e.g. abstract and introduction) were likely not sufficiently clear. As a consequence, in order to clarify the link between our results and the sensitivity of the POC to environmental change, as well as the connection between krill and CO₂ uptake, we made the following changes:

1-changed the final sentence in the abstract from “Since moulting rate is environmentally influenced, our results highlight the sensitivity of POC flux to rapid regional environmental change” to “Since exuviae production depends on temperature, our findings highlight the sensitivity of C flux to rapid regional environmental change”.

2-added the following sentence at the beginning of the introduction “Krill can have an important role in regulating the magnitude of carbon stored in the ocean via the biological pump (BCP) (i.e. the process that draws down atmospheric carbon dioxide (CO₂) through the fixation of inorganic carbon by photosynthesis and the consequent export and sequestration of carbon to the deep ocean) (Steinberg, et al. 2015)

3-modified the final paragraph in the introduction as follows: We found that, in the north Scotia Sea, krill exuviae can contribute to C flux at a similar magnitude to FPs. Sea-ice decline, ocean warming and other environmental stressors act in concert to modify the abundance, distribution and life cycle of krill (Schiermeier, 2010; Flores et al., 2012). Here, we show the rate of moulting, and release of C-rich exuviae into the environment, is an important contributor to C flux. This must be taken into account when assessing Southern ocean carbon budget and krill harvesting practices (Cavan et al. 2019).

4- added the following paragraph at the end of the discussion: Increasing water temperature in the Scotia Sea, as a result of climate change, will likely have a negative impact on krill growth and biomass (Hill et al. 2013, Kelin et al. 2018). Here, we show for the first time the crucial role of krill exuviae as a vector for C flux in the Southern Ocean, a region which contributes significantly to the global C export production (Schlitzer et al. 2002). Thus a potential decrease in krill biomass is likely to impact the marine biogeochemical cycles. Further, since the krill moult cycle (and in turn exuviae production) depends on temperature (Buchholz et al. 1991), our findings highlight the sensitivity of C flux to rapid regional environmental change (Schiermeier, 2010; Flores et al., 2012).

New references:

-Klein ES, Hill SL, Hinke JT, Phillips T, Watters GM (2018) Impacts of rising sea temperature on krill increase risks for predators in the Scotia Sea. PLoS ONE 13(1): e0191011.

<https://doi.org/10.1371/journal.pone.0191011>

-Schlitzer, R. Carbon export fluxes in the Southern Ocean: results from inverse modeling and comparison with satellite-based estimates. *Deep Sea Res. Part II Top. Stud. Oceanogr.* **49**, 1623–1644 (2002).

-Hill SL, Phillips T, Atkinson A. Potential climate change effects on the habitat of antarctic krill in the weddell quadrant of the southern ocean. *PLOS ONE.* 2013;8(8): e72246. pmid:23991072

- Steinberg, K. D. et al. Long-term (1993–2013) changes in macrozooplankton off the Western Antarctic Peninsula. *Deep Sea Res. I* 101, 54–70 (2015).

INTRODUCTION

R L23: Should it be krill “are” a fundamental conduit?

L24: Character issue throughout MS for benthic-pelagic coupling (as well as in references).

L45-46: Change to, “Chitin, a polysaccharide that can be completely remineralized to become a source of dissolved organic carbon, comprises 13% of exuvia.”

AU All the changed have been made as suggested.

R L67-68: It’s important to note that the Scotia Sea represents an important location for dissolved inorganic carbon drawdown whereas krill fecal pellets and exuviae would be contributing to organic carbon export. Please ensure to clarify this difference in the MS.

AU We clarified the difference in the sentence as followed “This study adds new insights on the important role of krill as a vector for C export in a region that contributes significantly to global atmospheric carbon uptake”.

R L69: What is “this” modifying? This carbon drawdown?

AU We clarified the sentence by using “our results” instead of “this”.

R L72-73: Again, I find this connection weak. The study doesn’t address the effect of environmental conditions (e.g., temperature) on moulting. Instead, the authors could highlight they quantify the differences in C flux seasonally.

AU-We have already addressed this issue in a previous comment.

RESULTS

R L81-83: Please identify the relative contribution of exuviae and FPs separately to this 99.2% total POC flux.

AU The relative contribution was 52% (FPs) and 47.2% (Exuviae). This information has been added in the text.

R L88: Add a % symbol next to 37.8.

AU Done.

R L99-100: Should this be flipped to minimum and maximum levels of biomass? I would expect maximum biomass to occur in austral summer but the way the sentence is currently stated makes it appear the maximum is in winter.

AU Yes, correct, winter –summer needed to be flipped. Thank you for spotting it.

R L103: Period after SM_1.

AU Done.

DISCUSSION

RL109: Contribution of krill “carcasses” can...

AU The sentence was referring to the total krill contribution so we decided to not add “carcasses”.

R L132-135: Tie this single sentence paragraph into the paragraph above it (L124-131).

AU Done as suggested.

R-L145-147: What about habitat partitioning? A recent study showed distributions of calyptope and furcilia larvae concentrate offshore from the Scotia Sea whereas juveniles strongly concentrate on the Scotia Sea shelf and these distributions can vary seasonally (see Perry et al. 2019 PloS ONE). This process may also aid to explain why no eggs or krill larvae were observed in the sediment trap used in this study (L165-166).

AU-We thank the reviewer for the suggestion of the paper from Perry et al. 2019. Unfortunately the paper from Perry is not relevant to this study site because there is not calyptopes and furcilia of Antarctic krill in the South Georgia shelf. Recruitment at South Georgia is not the same as at the Peninsula. Krill is delivered to South Georgia from the Peninsula (e.g. Murphy et al. 2007), this is not local recruitment. To clarify this point we added this sentence into the main text: “Neither eggs nor larval stages of krill were found in our sediment trap samples at any time of year indicating an absence of successful spawning in this region. We did observe a decrease in the size of exuviae within the sediment traps during autumn (March), indicating a recruitment of juvenile krill into the study region. These krill were juvenile (15+ mm) and are most likely to have originated from areas upstream of South Georgia, such as the Antarctic Peninsula and outlying islands (Murphy et al., 2007a, Fielding et al. 2014, Reid et al., 2010).”

R-L180-181 Please explain how this estimation was determined either in the methods or briefly here.

AU-We have now inserted an additional section in Supplementary Material SM2 detailing how this calculation was made. We further note there was a typographic error in the unit in the original submission.

METHODS

RL203: Please include the preservatives used in the supernatant (may fit best in first paragraph of methods).

AU The following sentence have been added “Each bottle contained a solution of 4% formalin mixed with 5g Sodium Tetraborate (BORAX) to arrest biological degradation during sample collection and to avoid carbonate dissolution”.

RL204-206: Change sentences to, “Prior to splitting, “swimmers” (i.e., zooplankton that can enter the receiving cups while alive) were carefully removed. Samples were first ...

L219: Should be i.e.,

L241: Undo capitalization of “The”.

L250: Change “through” to “by”.

L256: Should be i.e.,

L267: Change “through” to “by”.

L269: Period after 2018).

Au All the suggested changes above have been made.

FIGURE CAPTIONS

R-L513: Please state in the caption for Figure 3 that only select months are shown. Is there a reason not all months were included?

AU-In order to calculate the frequency of krill standard lengths, we can only use the months (sample bottles) where a significant number of exuviae were collected. Also, for an accurate measurement of lengths, only exuviae in a relatively good condition were selected (see also Supplementary data 1.1. for information about the number of exuviae selected). We added this information in the figure caption.

R L515-516: Change caption to, “Monthly mean biomass determined from an inverse model calculation based on exuviae caught in the sediment trap.”

AU Changed as suggested.

FIGURES

R-Figure 1. This may be a formatting issue converting the figures into a pdf but some of the bars in the error bars and barplots are cutoff. If these figure conversions are correct, please adjust the y-axes of figure 1 subplots to include the error bar at the peak of the POC flux (1a), the bar outline for months Oct. and Nov. (1b), and the top of the error bar for winter (1c). Also, in the Figure 1 figure caption it states standard deviations are shown for 1b & 1c but they are not included in the figure as they are in 1c. Finally, the legend text for carcasses in Figure 1b should be capitalized to be consistent with the capitalization used for FP and Exuv.

AU-We made the corrections as suggested by the reviewer.

R-Figure 2. While ascetically pleasing, I find the information conveyed in this conceptual diagram could more easily be stated in a table. If the authors added more complexity to the diagram (i.e., how the primary krill carbon vectors varied by season) then I think it would be justified to keep.

AU-We believe that the conceptual diagram can provide a better visualization to the wide readership of Nature Communications, which includes non-experts on this subject matter. However, we agree the seasonal component is a nice addition to the diagram and we modified it as suggested by the reviewer. Please see the modified Fig. 2 for details.

R SUPPLEMENTARY DATA Note: The title for both supplementary files are different than in the primary manuscript text.

AU Thank you for spotting that. We updated the title in the supplementary material.

REVIEWERS' COMMENTS

Reviewer #1 (Remarks to the Author):

Thank you for addressing all of my comments so thoroughly. I really did enjoy the manuscript and, as always, find the research coming out of this group fascinating.

I'd like to clarify that my intent on reproducing your equations was to confirm that they could be replicated by any other reader given the descriptions provided. I completely agree that it is not necessary to reproduce Tarling's previous paper, but appreciate that you've clarified that 5 days should be used instead of the number of days each trap was open. My initial interpretation of Tarling's previous paper was that the number of days (5 in that case) represented the number of incubation days over which moults had been observed. While no incubations were run in the present study, I had assumed the 30-day trap window represented that "incubation period". I think the additional clarity in the text helps to explain this now.

Reviewer #3 (Remarks to the Author):

This manuscript represents a well-written, useful study analyzing the contribution of Antarctic Krill, *Euphausia superba*, to particulate organic carbon export (POC) in the Scotia Sea of the Southern Ocean. The major findings of the study show that krill exuviae flux is of similar orders of magnitude to that of fecal pellet flux, which indicates previous studies only quantifying krill fecal pellet flux may underestimate the overall contribution of krill to POC flux. The novel aspect is that no study to date has quantified the contribution of krill exuviae to POC flux in the Southern Ocean. In addition, the study utilizes an inverse model to estimate krill population size based on measured exuviae collected in the sediment trap. These data presented in this study will be valuable to researchers interested in Antarctic krill zooplankton, and to those interested in quantifying zooplankton contributions to the biological pump in the Southern Ocean. This study will influence thinking by encouraging other research programs with sediment traps in the Southern Ocean to quantify fecal pellets and exuviae when determining krill contributions to POC flux.

The authors did a fine job incorporating my suggested edits. Increased information on the inverse model calculations is useful and aids in reproducibility of the work. The adjustments to the figures also greatly improves them, particularly for the schematic diagram (Figure 2). I do not have any major additional comments to provide. However, I would encourage the authors to better incorporate some of the shorter paragraphs (i.e., L37-41 and L56-61) in the introduction to increase readability. L37-41 appears to fit well with the introductory paragraph on the contribution of krill to the biological pump.

Patricia Thibodeau, University of Rhode Island

R3-I would encourage the authors to better incorporate some of the shorter paragraphs (i.e., L37-41 and L56-61) in the introduction to increase readability. L37-41 appears to fit well with the introductory paragraph on the contribution of krill to the biological pump.

AU-We thanks the reviewer for the all positive comments and to agree on how we included all the suggested edits. Concerning the last suggestion to better incorporate the shorter paragraphs, we think that the introduction is clear and we prefer to leave the text like it is.